# Pollen identification through convolutional neural networks: First application on a full fossil pollen sequence

**Médéric Durand** [1] *, **Jordan Paillard**[1], **Marie-Pier Ménard**[1], **Thomas Suranyi**[1,2],
**Pierre Grondin**[3], **Olivier Blarquez**[1]

1 Département de Géographie, Université de Montréal, Montréal, Québec, Canada, 2 Laboratoire Chrono-Environnement, UMR 6249 CNRS, Université de Franche-Comté, Besançon, France, 3 Direction de la recherche forestière, Ministère des Ressources naturelles et des Forêts, Québec City, Québec, Canada

* mederic.durand@umontreal.ca

**Data Availability Statement:** The data underlying the results presented in the study are available from the Open Science Network at https://osf.io/t2xns/ (DOI: 10.17605/OSF.IO/T2XNS) The model

## Abstract

The automation of pollen identification has seen vast improvements in the past years, with Convolutional Neural Networks coming out as the preferred tool to train models. Still, only a small portion of works published on the matter address the identification of fossil pollen. Fossil pollen is commonly extracted from organic sediment cores and are used by paleoecologists to reconstruct past environments, flora, vegetation, and their evolution through time. The automation of fossil pollen identification would allow paleoecologists to save both time and money while reducing bias and uncertainty. However, Convolutional Neural Networks require a large amount of data for training and databases of fossilized pollen are rare and often incomplete. Since machine learning models are usually trained using labelled fresh pollen associated with many different species, there exists a gap between the training data and target data. We propose a method for a large-scale fossil pollen identification workflow. Our proposed method employs an accelerated fossil pollen extraction protocol and Convolutional Neural Networks trained on the labelled fresh pollen of the species most commonly found in Northeastern American organic sediments. We first test our model on fresh pollen and then on a full fossil pollen sequence totalling 196,526 images. Our model achieved an average per class accuracy of 91.2% when tested against fresh pollen. However, we find that our model does not perform as well when tested on fossil data. While our model is over-confident in its predictions, the general abundance patterns remain consistent with the traditional palynologist IDs. Although not yet capable of accurately classifying a whole fossil pollen sequence, our model serves as a proof of concept towards creating a full large-scale identification workflow.

## 1 Introduction

The scientific field of palynology is the study of spores and pollen grains. Through the analysis of discriminating morphological features, palynologists can associate microscopic pollen to

is available on GitHub at https://github.com/meddur/pollen_ai Researchers are encouraged to look through and further improve on the content located in those repositories.

**Funding:** This research was financially supported by Mitacs (MD, OB; grant no. IT14197), the Nature and Technology Research Fund of Québec (FRQNT) (MD; grant no. 302964), the Études supérieures et postdoctorales of the Université de Montréal (ESP) through the artificial intelligence grant (MD), as well as the Ministère des Ressources naturelles et des Forêts du Québec (MD; PG). The funders had no role in study design, data collection and analysis, decision to publish, or preparation of the manuscript.

**Competing interests:** The authors have declared that no competing interests exist.

their parent species. For decades, palynology has been a useful method in numerous disciplines including honey production and allergen forecasting through the use of airborne pollen [1, 2]. Moreover, in the field of paleoecology, it has been of significant importance as a tool in the reconstruction of past environments and vegetation. The identification of fossilised pollen grains extracted from a sediment core allows paleoecologists to study past vegetation communities and their evolution through time [3]. Traditionally, this method involves a highly trained analyst identifying pollen under a light microscope. As between 300 and 500 pollen need to be identified within a sample for it to be statistically significant [3], palynology is generally understood to be a long and tedious process.

Achieving automation in palynology could solve many problems besetting the field–it could greatly accelerate the identification process while reducing the bias and uncertainty brought on by an analyst. This is because pollen grains are often morphologically similar, especially if belonging to closely related species. Significant effort has been spent towards improving taxonomic resolution, identification consistency between analysts, and identification accuracy for pollen with very similar morphology [4, 5]. That being said, expecting consistent results between different analysts should not necessarily be expected [6, 7]. Automating palynology would help guaranteeing a degree of consistency in both an analyst's work and between the work of different analysts. Furthermore, by decreasing analysis time, automating palynology could result in more samples being analysed. In the context of palyno-paleoecology, increasing the number of samples along a single sediment core would result in a higher temporal resolution. Analysing fossil pollen sequences at a high resolution would allow paleoecologists to infer ecological processes with more accuracy. In the hypothetical case of a species' pollen suddenly disappearing from pollen assemblages, a high temporal resolution analysis could help quantifying the species' rate of decline. Ideally, a classification algorithm would be combined with an automated slide scanner, which would allow for a large-scale detection and classification workflow of numerous images along a sediment core. Numerous studies show promise in combining deep learning and slide scanning [8–11].

In the past years, considerable effort has been spent automating pollen grain identification, mainly using deep learning algorithms [11–14]. Images of pollen grains are first captured either manually by cropping the pollen under a light microscope or automatically with the use of a slide scanner [8, 10, 11, 14]. Then, deep learning models are trained using those images in order to classify other similar looking pollen. For a time, the main focus of papers regarding this matter revolved around optimizing feature selection and extraction–generally understood as a model's capacity to pick out a pollen grain's important morphological features (size, texture, symmetry, number of pores, exine thickness, etc. . .) and to identify which of those features are discriminatory. The discriminatory features, once properly weighted, are then used to train a classifier. Numerous papers proposed such pollen classification models featuring varying degrees of user input in the feature selection and extraction phases [15–18]. In recent years, the advent of convolutional neural networks (CNNs), a type of deep learning network, has permitted the automatic extraction and selection of relevant features without human supervision [19]. This has brought on a number of very reliable pollen classification models [10, 11, 13, 14, 20–23], each of them tailored to their own needs.

Despite all of this work, there are only a few deep learning studies that trained a CNN that is adapted to the large-scale detection and classification of fossilised pollen. Recently, [11] has achieved a high degree of success on classifying fossil pollen images captured using a slide scanner. Their proposed workflow seems to be right on track to image and classify full fossil datasets. To our knowledge, there are two reasons that contribute to the scarcity of similar studies: i) most published deep learning algorithms are trained using clean pollen samples, usually fresh airborne pollen or extracted from honey. However, fossilised pollen are often

bent, broken, covered by debris or simply clumped together. They also rarely show sign of cellulose content, usually quite visible in fresh pollen. Since the training data must be representative of the target data, this is an issue. New methods that take into consideration the specificity of fossil pollen need to be integrated into the research. ii) palynologists that did train their algorithms using fossilised pollen, did so by manually cropping the pollen from a noisy background and labelling them, thus removing microscopic minerals and other non-pollen palynomorphs (NPPs) from the final images [12, 13, 23]. While manually cropping pollen avoids a problematic step in automated pollen recognition, it is time intensive and so is better suited for small-scale and precise classification problems, as seen in [14]. Large-scale automation implies the use of an automated slide scanner and the inevitable presence of NPPs, organic debris and images of large pollen clumps in the dataset.

This paper aims to train a CNN adapted to the classification of images of fossilised pollen found in Québec organic sediment cores and captured using an automatic slide scanner. We want our CNN (henceforth referred to as 'the model') to be adapted to a large-scale workflow of fossil pollen identification. Thus, we present a fossil pollen extraction method that aims to minimize the time spent per sample in the lab.

In order to train our model, we use fresh pollen. By allowing us to scan a full reference slide of any given pollen taxa, we are able to gather far more training images (Table 1). Using fossil pollen as training data would have implied picking out individual pollen grains from a slide, identifying them–possibly introducing analyst bias–and imaging them. This would have also meant that unidentifiable pollen (corroded, broken, or with distinctive features missing) would not have been included in the training data. This method is better suited for small-scale and precise classification problems that feature very similar pollen grains, as seen in [13, 24].

We choose to further evaluate our model by testing it on a fossil pollen image dataset (Table 1) (*i.e.* testing our model on data originating from a different source than the data used to train the model). This independent dataset is composed of fossilised pollen extracted from the sediment core of lac Bélanger, situated in central Québec (Fig 1). We resort to generating our own dataset because there exists no other public, significantly large, normalized dataset of fossilised pollen usually found in Québec organic sediments.

The fossil pollen grains were digitized using an automatic slide scanner. Using this fossil dataset, a first palynologist/model comparison is done using 20,151 unlabelled slide-scanner images distributed along 30 samples–assessing a model's accuracy compared to a human counterpart is best done using the same material [7]. We then compare the full high-resolution pollen diagram (created using 196,562 unlabelled images distributed among 271 samples from the full fossil image dataset) to a low temporal resolution pollen diagram created using traditional

**Table 1. Dataset glossary: This reference table indicates where the data comes from and what it is used for.** Each dataset falls under either the CNN dataset (Non-fossil images used to train, test, and validate the model) or the fossil image dataset (Fossil object images captured using the Classifynder automated slide-scanner; used to further test the CNNs).

| Dataset | Description | Labeled | # images | Relevant Methods |
|---|---|---|---|---|
| **CNN dataset** | Images used to train and test the CNNs; Either fresh pollen or from the ref. collection | Y | 16,331 | 2.2, 2.3 |
| **Training dataset** | Used to train the CNNs | Y | 12,248 | 2.2, 2.3 |
| **Testing dataset** | Used for the first round of testing (Fig 5) | Y | 2,450 | 2.2, 2.3 |
| **Validation dataset** | Used to validate training, to compute the negative loss likelihood metric (NLL) and to calibrate the CNNs | Y | 1,633 | 2.3 |
| **Full fossil image dataset** | Classifynder slide-scanner images taken from the lac Bélanger fossil sediment core; Used to plot the high-resolution model pollen diagram (Fig 8) | N | 196,562 | 2.4; 2.5 |
| **Fossil testing dataset** | 30 samples taken from the Fossil image dataset; The Classifynder slide-scanner images were visually labelled by a palynologist; Used for the second round of testing (Figs 6 and 7). | Y | 20,151 | 2.5 |

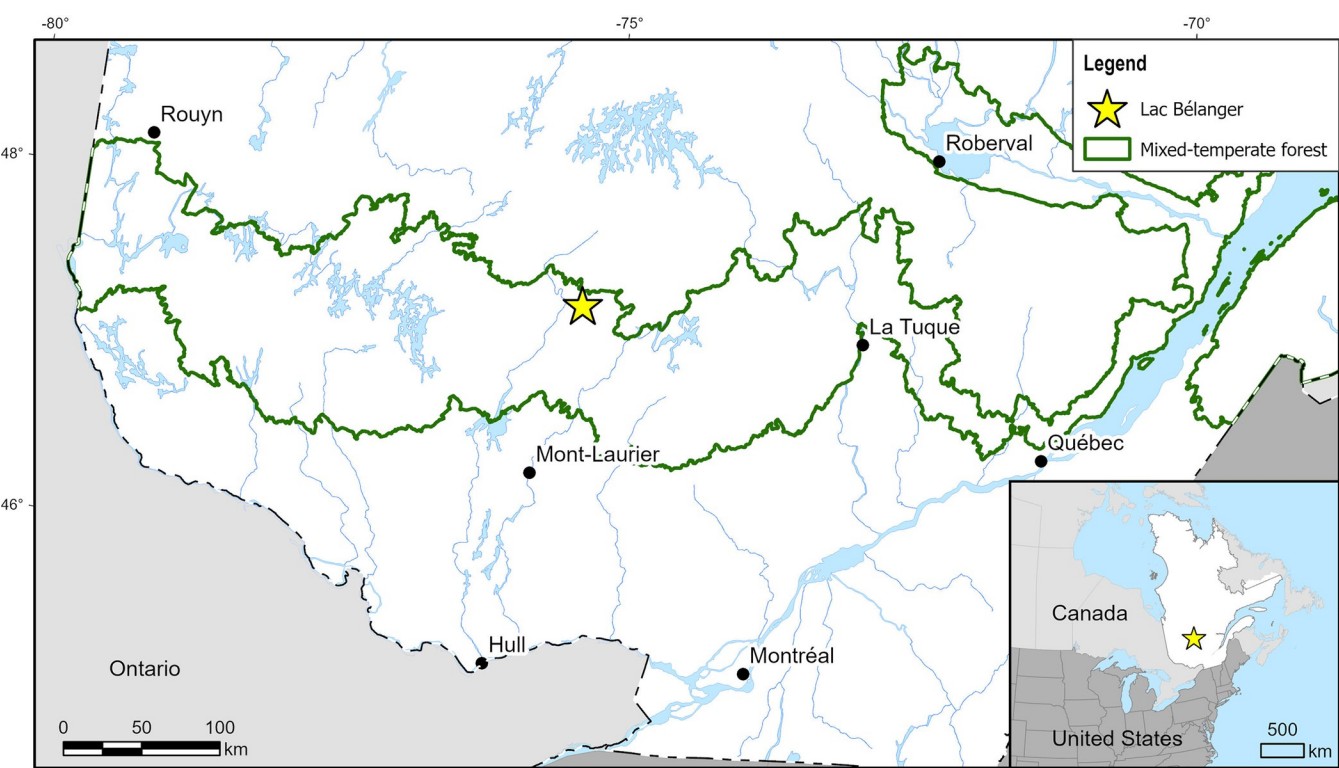

**Fig 1. Map of the study site.** Lac Bélanger, whose organic sediment core is used to generate the full fossil dataset. It is located within Québec's mixed-temperate forest, which is comprised of both temperate forest species (*Pinus strobus*, *Acer saccharum*, *Quercus*, *Betula alleghaniensis*) and stress-resistant boreal species (*P. banksiana*, *Abies balsamea*, *Picea*, *B. papyrifera*).

palynology. We hypothesize that while a drop in model accuracy might be observed, the model's predictions will resemble results obtained through traditional fossil palynology. We believe that the tendencies in pollen abundance observed between the two methods will be consistent with one another.

In this paper, we present the first application of an automated workflow on a full fossil pollen sequence. We put forward an accelerated fossil pollen extraction method that we believe is capable of outputting pollen samples suitable for an automatic slide-scanner in a fast and efficient manner. We identify the obstacles and pitfalls encountered gathering data. We compare the traditional and AI-powered methods using pollen diagrams in order to establish the limitations of our model.

## 2 Methods & material

### 2.1 General approach

The CNN dataset (the training, testing and validation data) is constructed using both reference slides and fresh pollen. Inspired by the multi-CNN method presented in [13], we train multiple small CNNs and link them hierarchically into a full network–the model (Fig 2). This model allows for a classification to be stopped at a higher taxonomical level. By doing so, an uncertain model prediction can result in a pollen being classified at the genus level rather than down to the species level. We choose to limit the number of different classes the model is trained on to 14. These include: the balsam fir (*Abies balsamea* [L.] Mill.), red maple (*Acer rubrum* L.), sugar maple (*Acer saccharum* Marsh.), mountain alder (*Alnus crispa* Aiton), grey alder (*Alnus rugosa*

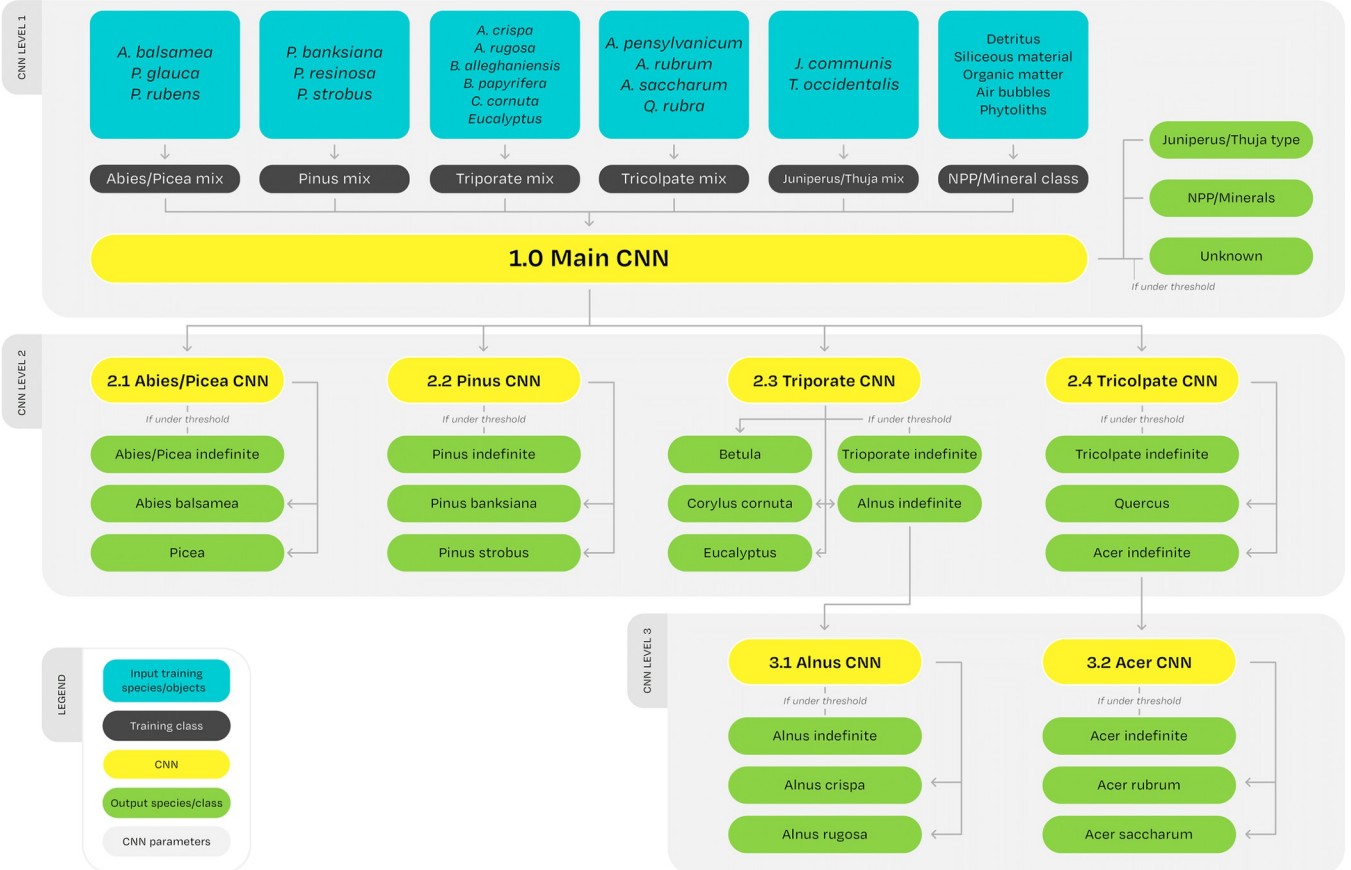

**Fig 2. Model flowchart.** It shows the composition of the training data (blue boxes that are grouped into the dark-gray boxes), how the seven different CNNs are hierarchically joined in a full network (yellow boxes and arrows) and their outputs (green boxes). In order to be classified, an unknown image travels top-down through the first CNN (Level 1; Main CNN) and then either be classified into an output class (green boxes) or go through a lower-level CNN (level 2) where the same process would repeat. An image could potentially travel down to the third level (Either Alnus CNN or Acer CNN), thus achieving a finer taxonomical identification.

L.), the birch genus (*Betula*), hazelnut (*Corylus cornuta* Marsh.), the eucalyptus genus, often used as exotic markers (*Eucalyptus*), the juniper-thuja morphological type class (*Juniperus communis* L. and *Thuja occidentalis* L.), the spruce genus (*Picea*), jack pine (*Pinus banksiana* Lamb.), white pine (*Pinus strobus* L.), the oak genus (*Quercus*) and an NPP/minerals class (Fig 2). Including the intermediate hierarchical classes, an image can be classified in one of 23 classes. After testing the model on the CNN training dataset, we test the algorithm using the lac Bélanger's fossil testing dataset.

## 2.2 Datasets and image acquisition

We assembled a complete dataset consisting of 16,331 images of pollen grains belonging to 18 different taxa (classes) usually found within North-Eastern American Holocene organic sediment cores–the CNN dataset. As is usually the case in traditional palynology, certain classes whose pollen are morphologically similar were combined into a single class, bringing down the total amount of pollen classes the algorithm was trained on to 13 (Table 2).

The pollen grains composing the CNN dataset originated from different sources. Some were found in the Université de Montréal's pollen refence collection. The pollen from classes that were absent from the collection were instead harvested directly from tree flowers and

**Table 2. Information on the training dataset.** Each pollen taxa (Species) makes part of a Class, usually made up of pollen of a similar morphological group. For each taxa, the source, their additional laboratory treatment (if any) and their number of grains is tallied. The training dataset is fed into the model in order to train it (Fig 2).

| Morphological type | Species | Class | Source | Additional lab treatment | Total grains |
|---|---|---|---|---|---|
| Aporate | *Juniperus communis* L. | Juniperus/Thuja type | Reference collection | | 1301 |
| Aporate | *Thuja occidentalis* L. | Juniperus/Thuja type | Fresh | | 194 |
| Tricolpate | *Acer pensylvanicum* L. | Tricolpate mix; Acer | Reference collection | Acetolysis | 251 |
| Tricolpate | *Acer rubrum* L. | Acer; Acer rubrum | Fresh | Acetolysis | 1272 |
| Tricolpate | *Acer saccharum* Marsh. | Acer; Acer saccharum | Fresh | Acetolysis | 302 |
| Tricolpate | *Quercus rubra* L. | Tricolpate mix; Quercus | Reference collection | | 1856 |
| Triporate | *Betula alleghaniensis* Britt. | Triporate mix; Betula | Fresh | Enzymes | 295 |
| Triporate | *Betula papyrifera* Marsh. | Triporate mix; Betula | Reference collection | | 675 |
| Triporate | *Corylus cornuta* Marsh. | Triporate mix; Corylus cornuta | Fresh | Acetolysis | 1777 |
| Triporate | *Eucalyptus* sp. | Triporate mix; Eucalyptus | Reference collection | | 1531 |
| Triporate* | *Alnus crispa* Aiton | Triporate mix; Alnus; Alnus crispa | Fresh | | 773 |
| Triporate* | *Alnus rugosa* L. | Triporate mix; Alnus; Alnus rugosa | Fresh | | 1931 |
| Vesiculate | *Abies balsamea* (L.) Mill. | Abies/Picea mix; Abies balsamea | Reference collection | | 1183 |
| Vesiculate | *Picea glauca* (Moench) Voss | Abies/Picea mix; Picea | Fresh | Enzymes | 344 |
| Vesiculate | *Picea rubens* Sarg. | Abies/Picea mix; Picea | Fresh | Enzymes | 532 |
| Vesiculate | *Pinus banksiana* Lamb. | Pinus mix; Pinus banksiana | Reference collection | 50% enzymes; 50% acetolysis | 923 |
| Vesiculate | *Pinus resinosa* Aiton | Pinus mix; Pinus banksiana | Reference collection | | 269 |
| Vesiculate | *Pinus strobus* L. | Pinus mix; Pinus strobus | Fresh | Enzymes | 922 |

*While of stephanoporate morphology, Alnus pollen have been assigned to the triporate morphological group since they are of similar size and shape, save for the amount of pores. This emerged as a more pragmatic solution than introducing a new, distinct class for stephanoporate grains.

N.B.: Throughout this paper, species names are italicized while class names are not. This is done in order to help distinguish the two.

cones during each species' flowering period (Table 2). They then were stored in 70% proof ethanol tubes until further treatment. The dispersing effect of ethanol accentuated the separation of the pollen grains from the flowers and cones.

Each sample was then submerged in a potassium hydroxide solution (KOH 20%) under heat for 20 minutes before being washed with water and filtered through a 150 μm mesh. When put under a light microscope, certain species' pollen grains still showed visible cellulose content. In order to clear the cellulose contained within the grains, two different methods were used. We followed an enzyme-based method to successfully clean out the grains of *B. alleghaniensis*, *Picea glauca*, *P. rubens* and *P. banksiana* [25]. This was achieved by using an equal part pectinase to cellulase ratio (0.1 g), 10 ml distilled water and a 10 ml citrate buffer, bringing the final pH to 6.5. The remaining species' grains still containing cellulose were cleaned through acetolysis (Table 1).

Prior to the pollen solution being mounted on the slides, the 70% proof ethanol was emptied from the tubes and replaced by 90% proof ethanol in order to accelerate evaporation. The slides were then mounted using a warm glycerin jelly solution and given a few minutes for the ethanol to evaporate. The slides were then sealed and cooled off in order to let the solution and pollen set.

The images were acquired using Veritaxa's Classyfinder, an automated slide scanner, at a 40x magnification under dark-field illumination. The pollen was automatically identified through the use of morphological criteria input by the user (see S1 File). Using the Classyfinder's software, the final images were generated as a Z-stack, fusing the different images of the grain taken along the depth at a 1(one) μm interval [8]. The final image data are a single

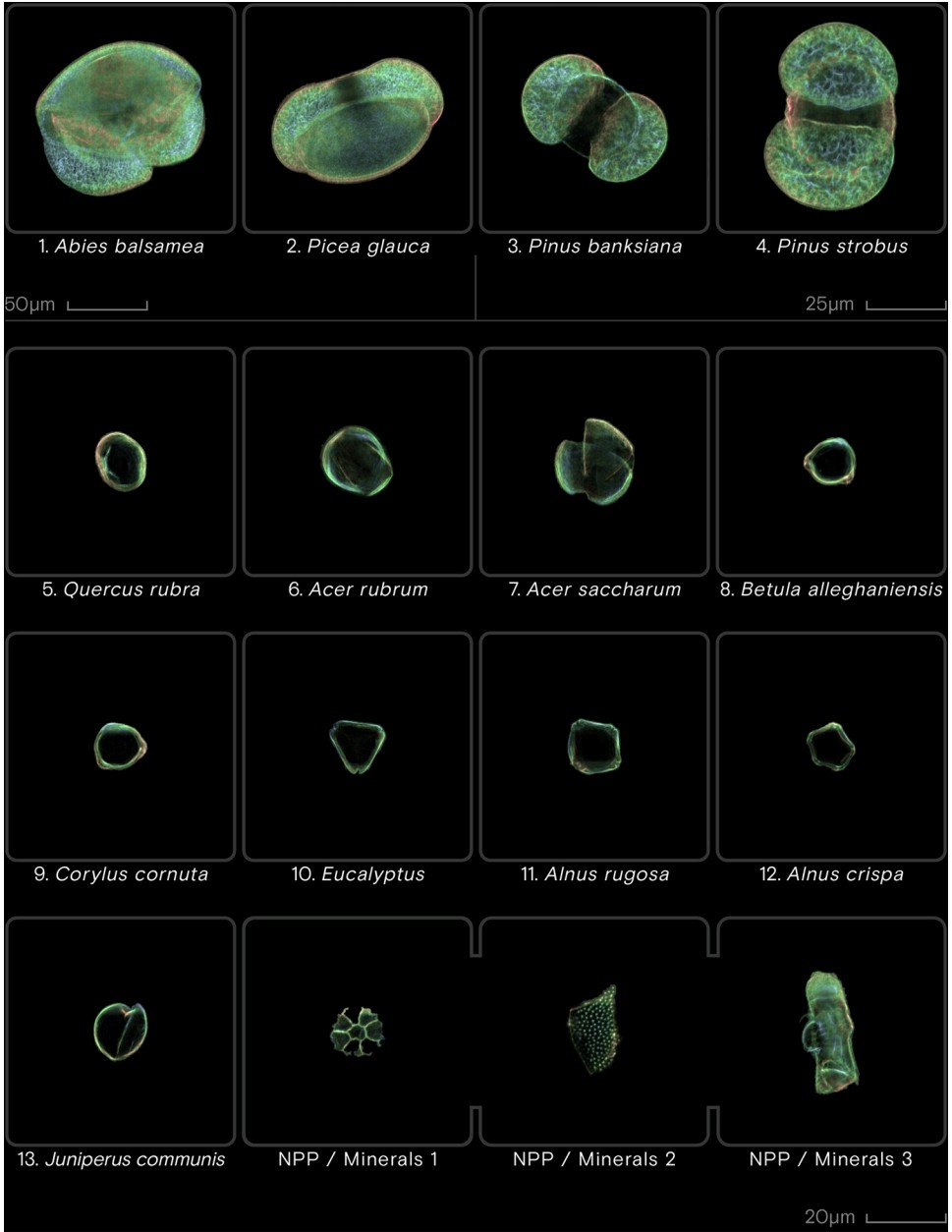

**Fig 3. Different pollen taxa as captured by the Classyfinder slide scanner.** Note the change in scale between images 1–2, 3–4, and the rest.

image, fused from the Z-stack (Fig 3). Using the same image capture method, a 14th class was added to the CNN dataset, comprising of 1,375 images of various NPP/minerals that can be found in lake sediment pollen slides, namely clay particles, chironomid, phytoliths and air bubbles, all mistakenly captured by the Classyfinder microscope. The CNN dataset was then transformed to grayscale and rescaled to 128x128 pixels. 75% of the CNN dataset was randomly selected to create the training dataset. The testing dataset was constructed using 15% of the CNN dataset, while the validation dataset was formed using the remaining 10%.

**Table 3. CNN training parameters.** This table synthesises the parameters used in the training of the seven CNNs. The "Images per class" parameter accounts for all of the corresponding training, testing and validation images. The "Under threshold" metric corresponds to the percentage of test images that fell under the 0.7 confidence threshold during evaluation i.e. the proportion of the test set that was classified as "unknown" (μ: 4%). The NLL Improvement metric (Negative log likelihood) quantifies the impact of Temperature-scaling using the validation data.

| ID | CNN name | Images per class | Epochs | Accuracy | Architecture | NLL Improvement | Under threshold |
|---|---|---|---|---|---|---|---|
| 1.0 | Main CNN | 1,200 | 200 | 95.1% | Deep | 1.309 -> 0.219 | 3.3% |
| 2.1 | Abies/Picea CNN | 1,000 | 250 | 92.6% | Deep | 0.462 -> 0.229 | 3.0% |
| 2.2 | Pinus CNN | 830 | 200 | 89.6% | Deep | 0.49 -> 0.281 | 9.2% |
| 2.3 | Triporate CNN | 110 | 600 | 90.5% | Deep | 1.002 -> 0.395 | 3.2% |
| 2.4 | Tricolpate CNN | 720 | 550 | 83.3% | Deep | 0.527 ->0.412 | 6.0% |
| 3.1 | Alnus CNN | 575 | 170 | 96.5% | Shallow | 0.445 -> 0.09 | 2.8% |
| 3.2 | Acer CNN | 310 | 280 | 90.9% | Deep | 0.483 -> 0.301 | 2.2% |

## 2.3 Model conception and image classification

The deep learning project was coded using Python. The training was done using the CNN training dataset (Table 1), employing both the Keras and Tensorflow APIs, although the saved models can be loaded into a PyTorch framework. Following the method found in [13], multiple CNNs were trained and hierarchically assembled, forming a "multi-CNN" system. Each of those seven CNN was trained on its own subset of pollen classes according to their morphological similarity (Fig 2). When fed into the system, the input data are either classified as belonging to a particular pollen class, sent to a CNN of a higher hierarchical level, or, if their prediction fails to reach a confidence threshold, are dropped out of the classification process. Drop-outs remain accounted for and are registered as belonging to their parent class, *e.g.*, having reached the Acer CNN, an *A. saccharum* pollen failing to clear the confidence threshold would still be counted as belonging to the *Acer* genus, as a real palynologist would. Considering that morphologically similar pollen grains generally belong to taxonomically related species, the class composition and total amount of CNNs has been determined through morphological similarity.

Since the training of a deep learning algorithm using imbalanced data may result in the overfitting of one of its classes, each CNN's corresponding dataset size was limited to the number of images present in the least populated class (Table 3). As is commonly done for the training of image classification models, our model used data augmentation techniques during training. This module allowed the simultaneous generation of 'new' training images during each iteration (batch sent through the network), *de facto* increasing the total amount of training data available for each CNN. These 'new images' were generated by randomly applying rotations (0–45˚), shearing (0–15%) and lowering luminosity (0–20%).

The CNNs' common core architecture is comprised of 6 trained layers (Fig 4). The main elements of a CNN are present: two convolutional layers with a ReLU activation function are followed by a single max pooling layer, doubling the filter size and connecting to two additional convolutional layers until the fully connected layer is reached at the top. For a relatively recent review of deep learning concepts and terms, see [19].

In the intial layers of each model, two convolutional layers and a max pooling layer were imported from the VGG16 pre-trained image classification network [26]. VGG16 was trained on the ImageNet dataset, which at the time contained more than 14 million labelled images separated into 22,000 distinct categories. This process, called transfer learning, is employed to make use of another network's pre-trained layers in order to increase the feature extraction capacity of our own models. The layers imported from the VGG16 neural network were taken from its first block, situated at the bottom, wherein the layers were trained to extract basic

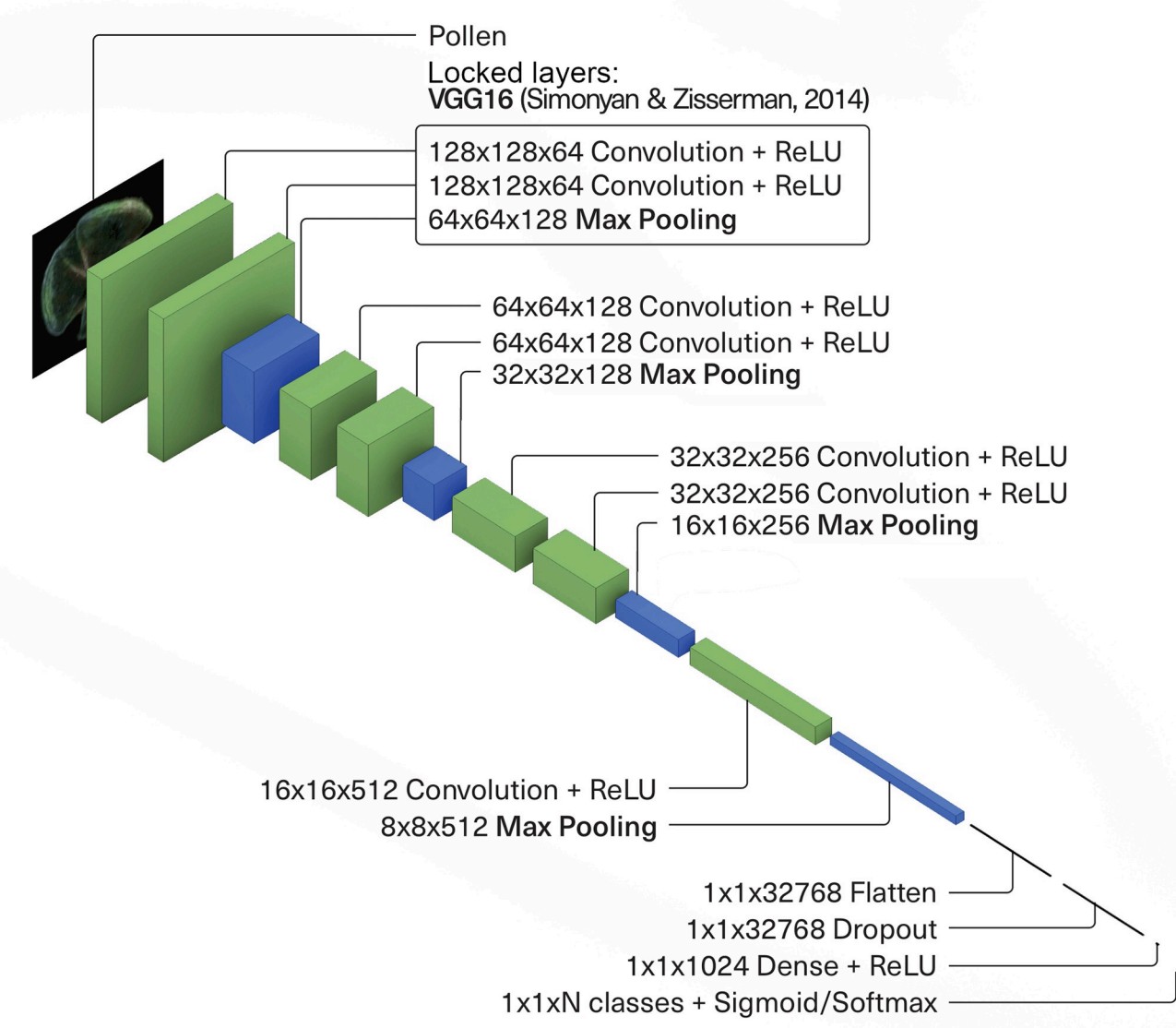

**Fig 4. Architecture of the convolutional neural networks.** The first three layers present after the inputs (left to right) are transfer learning layers. The next six layers represent the common core, present in all seven networks. The following two layers were present only in the deep architecture networks. For the Alnus CNN, a shallower model was used wherein the last convolutional and max pooling layers are removed. The fully connected layer is comprised of the last four layers. See [19] for a review on the terms and layers discussed in this paper.

features. Using those already trained layers gives our CNNs a "headstart" in the initial training steps. Relying on transfer learning instead of our own models' low level randomized feature extraction allows for deeper feature processing in our models' subsequent convolutional layers. This technique was shown to be effective in palynological image classification problems [20, 23] and is discussed in depth in [19]. A dropout layer was inserted as the beginning of the fully connected layer, dropping 50% of each batch's tensors, lowering the overfitting chances in the process. The fully connected layer is ultimately activated using a softmax activation function.

In accordance with [27], the softmax output tensors–the predictions–were transformed using temperature scaling, a variant of Platt scaling. By scaling the output logits by the parameter $T$ ($T$>0), temperature scaling calibrates the models' outputs by lowering the prediction

confidence without affecting the models' accuracy. Employing the validation dataset images, each model's *T* was calculated using the negative log likelihood (NLL) and the validation loss metric prior to predicting. A 0.7 threshold was applied to the prediction outputs. Each image whose prediction fell under this threshold was assigned the *Unknown* label and dropped out of the model. If the image reached a classification level higher than 1 before falling under the threshold, it was instead marked as an indefinite object belonging to its parent class before dropping out of the model.

Other than dataset size and temperature *T*, using a multi-CNN approach has allowed us to tailor fit certain parameters according to the available data. Namely, each model was trained using its own optimized number of epochs (Table 3). Furthermore, two different model architectures were used to train the seven CNNs: a shallow and a deep architecture (the latter can be visualized on Fig 4). Both architectures share the same common core, batch size and input size. The shallow architecture was only used to train the Alnus CNN (Table 3), since its input data tended to overfit rapidly. Used to train the remaining six CNNs, the deeper architecture adds both a single additional convolutional layer and a max pooling layer, thus doubling the number of output filters in the fully connected layer. As with each model's chosen number of epochs, the various parameters and their optimal values, including the choice of architecture, *T*, batch size and transfer learning layers, all depend on the dataset size and overall image complexity and thus have been determined experimentally.

## 2.4 Fossil pollen extraction

In order to steer training accuracy and to optimize parameter tuning, the models have been tested using data from the testing dataset. As shown in section 2.2, the testing dataset was generated using fresh pollen, similarly to the training dataset. To further test the model, a fossil image dataset was created using fossil pollen extracted from lac Bélanger's sediments. The purpose of this second round of testing is to effectively test the model using images of fossilized pollen which have undergone various degrees of weathering and decay.

Lac Bélanger is located in central Québec (47.476045, -75.177318) and was sampled in the summer of 2017 (Fig 1). With an annual average temperature close to 0˚C and annual precipitation averaging 1,000mm, it is situated within Québec's mixed-temperate forest. While regionally *Abies balsamea* and *Betula alleghaniensis* usually make up the dominating stands, lac Bélanger is characterized by a bordering *Acer saccharum* stand, where it reaches its northern distribution limit. Its extracted sediment core measures 8.1 meters. 10 samples have been dated using radiocarbon dating techniques. Its basal age is 9,978 (±290) calibrated years before present (cal. BP). Its temporal resolution averages 12.4 years/cm.

271 levels were chosen and sampled along the sediment core's depth at an interval averaging 3 cm. Pollen composing the fossil image dataset have been extracted from the lacustrine sediment samples using a different method than those extracted in section 2.2. The traditional method [28] relies on acidic treatments to dissolve leftover cellular content (acetolysis) and siliceous material (hydrofluoric acid), with the unintended side effect of further damaging the pollen grains. Moreover, only a small portion of the CNN training dataset has undergone acetolysis. While a human palynologist would see little to no problem discriminating pollen from NPP/minerals, especially siliceous, dispensing with the removal of such content would put an unnecessary strain on both the Classyfinder software and the model. Furthermore, since the usage of an automated pollen identification method is hugely beneficial to large scale work, minimizing the time spent per sample is crucial. Considering these constraints, the development and usage of a new method was necessary.

One (1) cm$^3$ was sampled from each sediment sample into a 10ml tube. A known quantity of an exotic marker solution of *Eucalyptus* (15,319 ± 1,975 per cm$^3$) was added to each tube. Exotic pollen markers are used in palynology to calculate pollen influx (n/cm$^2$/year) and pollen concentration (n grains/cm$^3$). Consequently, it is necessary for *Eucalyptus* pollen to be in the training data. Each tube was then filled with a potassium hydroxide solution (KOH 20%) and immersed in an ultrasonic bath filled with boiling water for 20 minutes before being washed with distilled water. This has the effect of removing humic acids and deflocculating the pollen grains from themselves and other organic particles. Each pollen was then filtered through a 150 μm mesh and a 15 μm mesh. The content that had not filtered through the 15 μm mesh was then collected, put in a 10 ml tube and submerged in a heavy liquid of 1.7 density (sodium polytungstate). This method allows for a density based separation of the pollen solution from the siliceous particulates [29]. It has also been shown to be effective with gyttja and other organic-rich sediment [30]. After mixing the tubes' content, they were centrifuged twice in a row. The first centrifugation ran at 1,500 RPM for a duration of 5 minutes, while the second ran at 2,000 RPM for the same duration. The process results in a segmentation of the 'traffic' within the tubes: the lower speed centrifugation causes the less dense organic matter to float up while the faster centrifugation ultimately brings the denser siliceous material downwards. Using a pipette, the floating pollen were then collected and filtered through the 15 μm mesh. The material that did not filter through the mesh was then gathered and stored in 70% proof ethanol until further treatment. This method allows for the treatment of up to 30 samples in an 8-hour day.

## 2.5 Fossil pollen testing dataset

The 271 pollen slides were then mounted and the images captured similarly to the ones in section 2.3. These 196,526 unlabelled images form the fossil image dataset and were plotted as a full pollen diagram featuring pollen biozones computed using a statistical cluster analysis zonation method (CONISS). The broken-stick method [31] was used to identify the right number of significant zones to be adopted. 30 samples were chosen from the fossil image dataset at an interval averaging 30 cm, forming the fossil testing dataset. This dataset's 20,151 slide-scanner images were visually classified–but not individually labelled–by a human palynologist into the same output classes offered by our model, with the exception of *Alnus* images: since the image quality was too low, images visually classified as *Alnus* were not further classified into either the Alnus crispa or Alnus rugosa class. Pollen that were not present in the model (*e.g. Larix laricina* [Du Roi], *Tsuga canadensis* [L.]) were instead classified as *Unknown*. This was also the case with pollen that could not be properly identified because of the low image quality. Triporate or tricolpate pollen that were not used to train the model were still classified as such (e.g. *Fagus grandifolia* [Ehrh.] were classified as belonging to the Tricolpate class).

Identifying two-dimensional Z-stack images of pollen considerably limits the potential for both precision and certainty than doing so through a light microscope. That being said, we believe that this approach has resulted in consistent identifications and that it was the only way to isolate the model's "own" error–by bypassing any bias and errors introduced by the Classyfinder automated microscope during image capture.

# 3 Results

## 3.1 Training results

Our model has a 91.2% average per-class accuracy (APC) (Fig 5). The top-level classifier has a 95.1% APC. The mid-level classifiers show lower accuracies: the Abies/Picea CNN yields

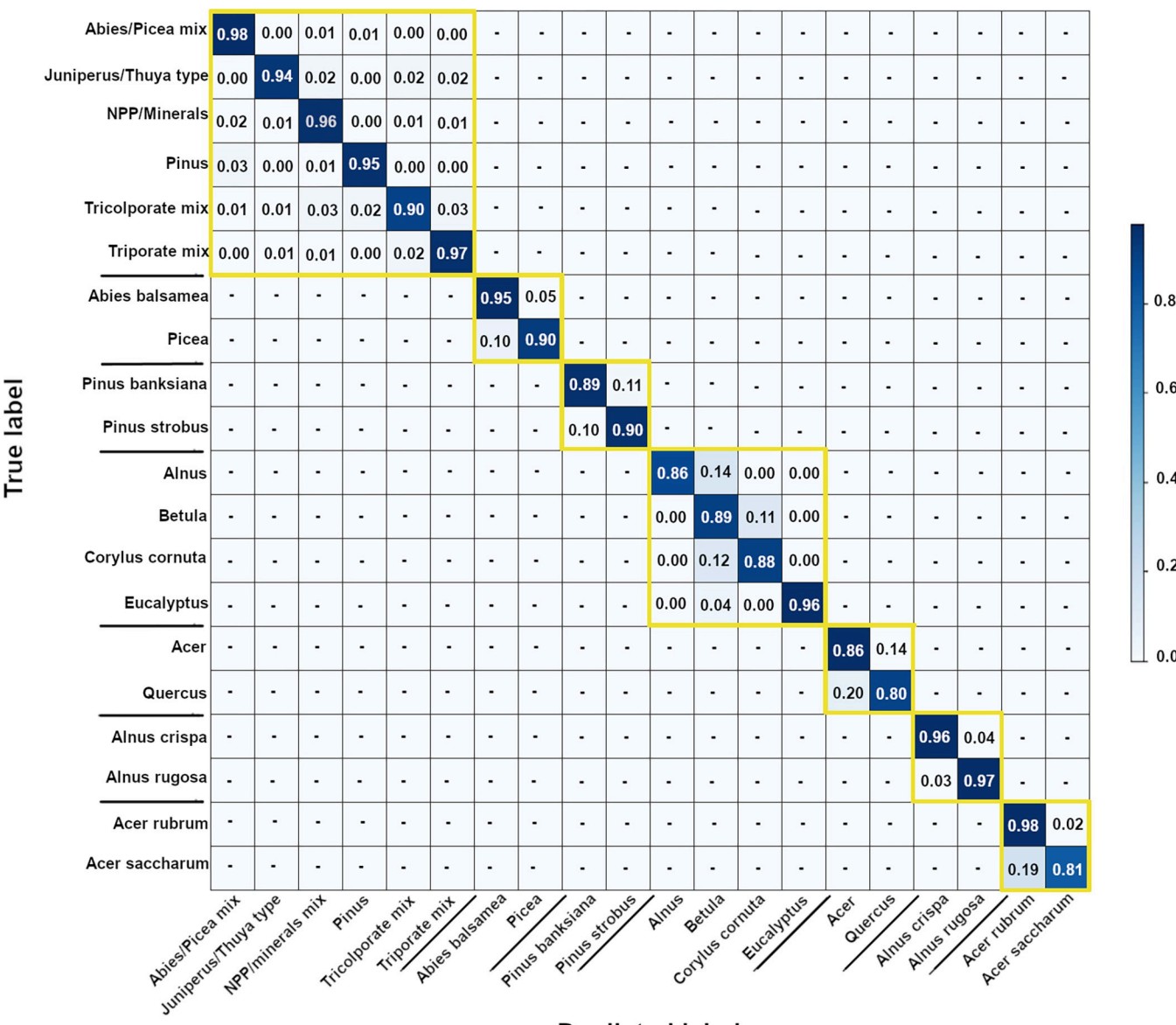

**Fig 5. The confusion matrix showing the training accuracy of our seven models when tested on the test dataset.** The models are arranged in hierarchical order from the top-left towards the bottom-right. The data has been normalized.

92.6% APC; the Pinus CNN 89.6% APC; the Triporate and Tricolpate CNN yield 90.5% and 83.3%, respectively. At the third taxonomic level, the Acer CNN has an APC of 90.9%, while the Alnus CNN tests at a 96.5% APC. The average proportion of test images whose prediction fell under the 0.7 threshold is 4%. Using temperature scaling, all the models have been calibrated. This is visible in the decreasing NLL values shown on Table 3, showing both pre- and post-calibration NLL values. The models that benefited the most from calibration were the two non-binary CNNs.

## 3.2 Comparison with the fossil testing dataset

We used a set of images independent from the training process to evaluate the model's accuracy. These images come from the fossil testing dataset and were labelled by a palynologist without the use of a light microscope (*i.e.* the palynologist used the Classyfinder slide scanner's two-dimensional Z-stack images). These predictions were then compared to the model's predictions (Fig 6). Scatterplots and their respective linear correlation curves were plotted in order to illustrate each class's correlation to the fossil testing dataset (Fig 7).

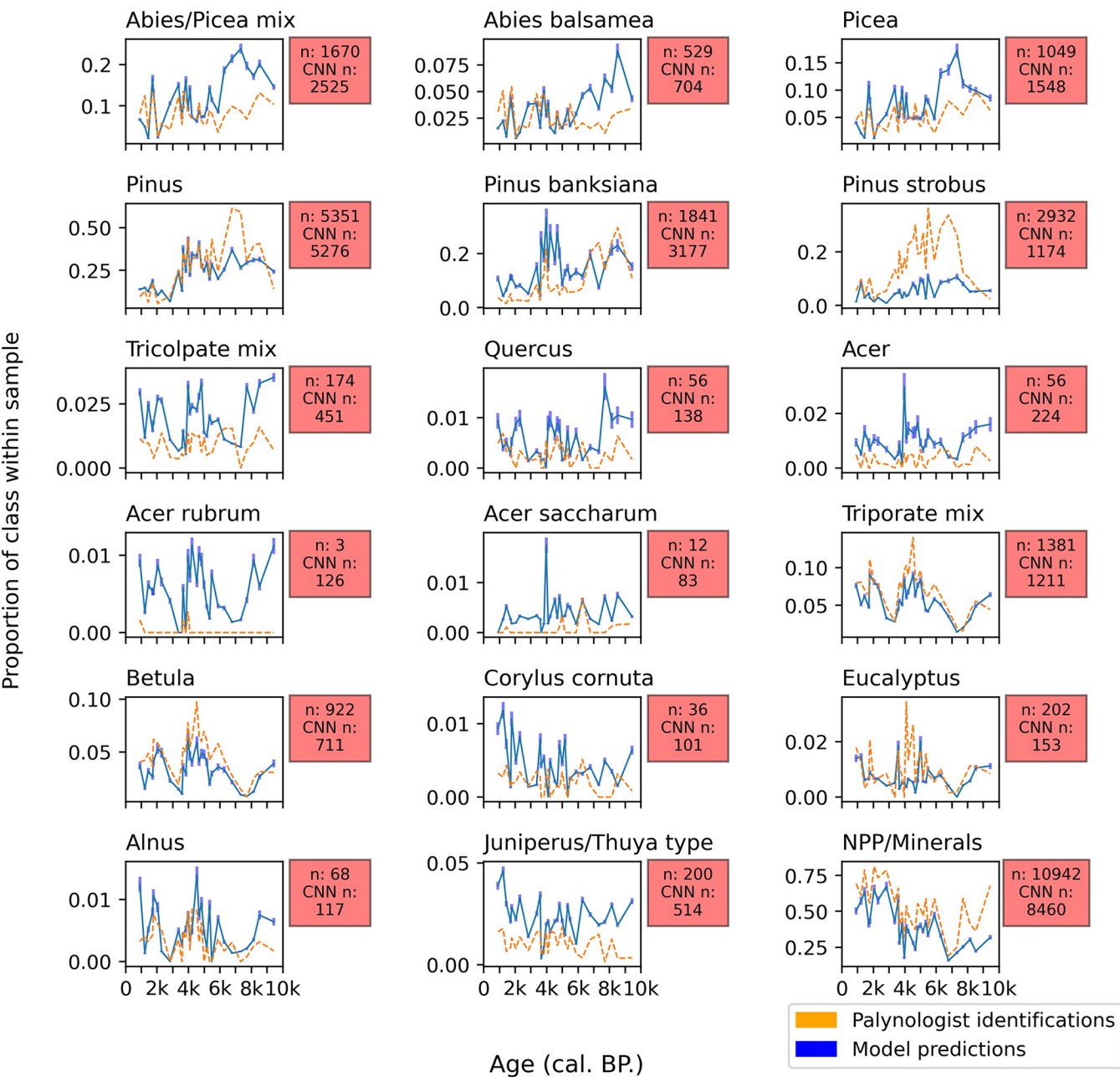

**Fig 6. Proportion over time of each possible output class, minus Alnus crispa and rugosa.** Both the observed n and CNN predicted n are next to their class plot. The time series corresponds to 30 carbon-dated samples analysed along lac Bélanger sediment core.

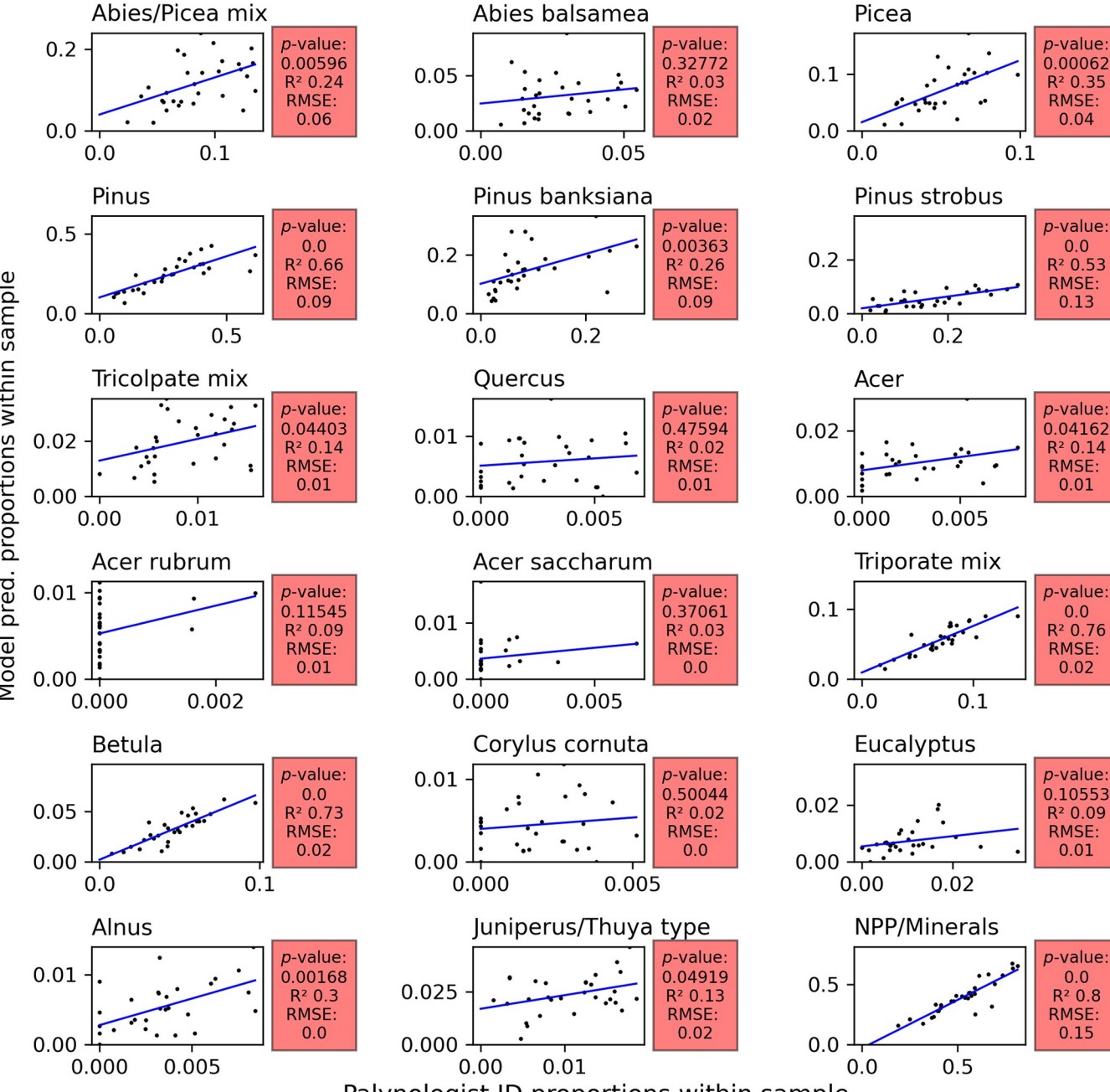

**Fig 7. Scatter plot of each possible output class, minus Alnus crispa and rugosa.** P-values under 0.05 indicate a correlation according to Student's t-test. The $R^2$ value indicates the goodness of fit of the regression curve, while the root-mean-square deviation (RMSE) indicates the spread of the data, i.e. if the goodness of fit is affected by outliers.

An average of 14.5% of the model's predictions fell under the 0.7 threshold and so were classified by the model as *Unknown*. In comparison, 7% of the images were classified as such by the palynologist. Compared to the CNN test set, this number falls to 4% (Table 3). This indicates that the calibrated model is half as confident as a palynologist when pitted against real

**Table 4. The results of the statistical tests are listed here.** The class column indent shows the classification level and respective parent classes (if any) of the 18 output classes. n refers to the number of grains identified by the palynologist, while n(CNN) refers to the number of grains as identified by the model. The average per-class accuracy (APC) has only been computed for non-final output classes. 6 out of 18 classes failed Student's t-test.

| Class | Classification level | n | n (CNN) | Average per class accuracy (%) of non final class | P-Value (α = 0.05) | $R^2$ | RMSE |
|---|---|---|---|---|---|---|---|
| **Abies/Picea mix** | 1 | 1670 | 2525 | 92.6 | 0.00596 | 0.24 | 0.06 |
| **Abies balsamea** | 2 | 529 | 704 | NA | 0.33 | 0.03 | 0.02 |
| **Picea** | 2 | 1049 | 1548 | NA | 0.00062 | 0.35 | 0.04 |
| **Pinus** | 1 | 5351 | 5276 | 89.6 | $<1\times10-5$ | 0.66 | 0.09 |
| **Pinus banksiana** | 2 | 1841 | 3177 | NA | 0.00363 | 0.26 | 0.09 |
| **Pinus strobus** | 2 | 2932 | 1174 | NA | $<1\times10-5$ | 0.53 | 0.13 |
| **Tricolpate mix** | 1 | 174 | 451 | 83.3 | 0.04403 | 0.14 | 0.01 |
| **Quercus** | 2 | 56 | 138 | NA | 0.48 | 0.02 | 0.01 |
| **Acer** | 2 | 56 | 224 | 90.9 | 0.04162 | 0.14 | 0.01 |
| **Acer rubrum** | 3 | 3 | 126 | NA | 0.11 | 0.09 | 0.01 |
| **Acer saccharum** | 3 | 12 | 83 | NA | 0.37 | 0.03 | $<1\times10-2$ |
| **Triporate mix** | 1 | 1381 | 1211 | 90.5 | $<1\times10-5$ | 0.76 | 0.02 |
| **Betula** | 2 | 922 | 711 | NA | $<1\times10-5$ | 0.73 | 0.02 |
| **Corylus cornuta** | 2 | 36 | 101 | NA | 0.5 | 0.02 | $<1\times10-2$ |
| **Eucalyptus** | 2 | 202 | 153 | NA | 0.11 | 0.09 | 0.01 |
| **Alnus** | 2 | 68 | 117 | 96.5 | 0.00168 | 0.3 | $<1\times10-2$ |
| **Juniperus/Thuja type** | 1 | 200 | 514 | NA | 0.04919 | 0.13 | 0.02 |
| **NPP/Minerals** | 1 | 10942 | 8460 | NA | $<1\times10-5$ | 0.8 | 0.15 |

world data. When predicting on the non-fossil data, the model is more confident than the palynologist.

Linear regression t-tests have been computed and plotted in order to test for statistical correlation between the two prediction sets. The statistical evaluation results are listed on Table 4. The resulting *P*-values indicate that 12 classes out of 18 show statistical correlation (α = 0.05). The classes that failed Student's t-test are Abies ($R^2$ = 0.03; *p*-value = 0.33; RMSE = 0.02), Quercus (0.02; 0.48; 0.01), both Acer rubrum (0.09; 0.11; 0.01) and saccharum (0.03; 0.37; <0.01), Corylus (0.02; 0.5; <0.01) and Eucalyptus (0.09; 0.11; 0.01). None of these classes were situated at the first level of the classification process. Apart from Abies and Eucalyptus, all the classes that failed Student's t-test have an n<60. A low n also correlates with a low $R^2$ (*p*-value<0.005). This is a limitation inherent to the R-squared metric.

Some palynologist predictions do not fall within the error margin of the model predictions (Fig 6). Specifically, plotting the two prediction sets alongside the time axis indicates that predictions tend to differ in older samples. The R-squared results tend to indicate varying regression model performance. Abies predictions show low accuracy, especially in samples older than 6,000 cal. BP, where the model seems to be over predicting. This pattern is also visible in Picea predictions. This coincides with Pinus which shows a pattern of underprediction, particularly in samples 6,000 cal. BP and older.

While Pinus strobus ($R^2$ = 0.53; RMSE = 0.13) predictions have a relatively high goodness of fit in its linear regression model (Fig 7), a pattern of underprediction is clearly visible along the time sequence (Fig 6). Although the NPP/minerals class ($R^2$ = 0.8; RMSE = 0.15) shows only a small tendency towards underprediction, along with Pinus strobus, there are only two classes out of eighteen that display a clear underprediction pattern. It could be inferred that the model is overconfident in its predictions.

Out of all the first-level classes, the Tricolpate ($R^2$ = 0.14; RMSE = 0.01) class shows the lowest performance, although it does pass Student's t-test. This is further noticeable in its children

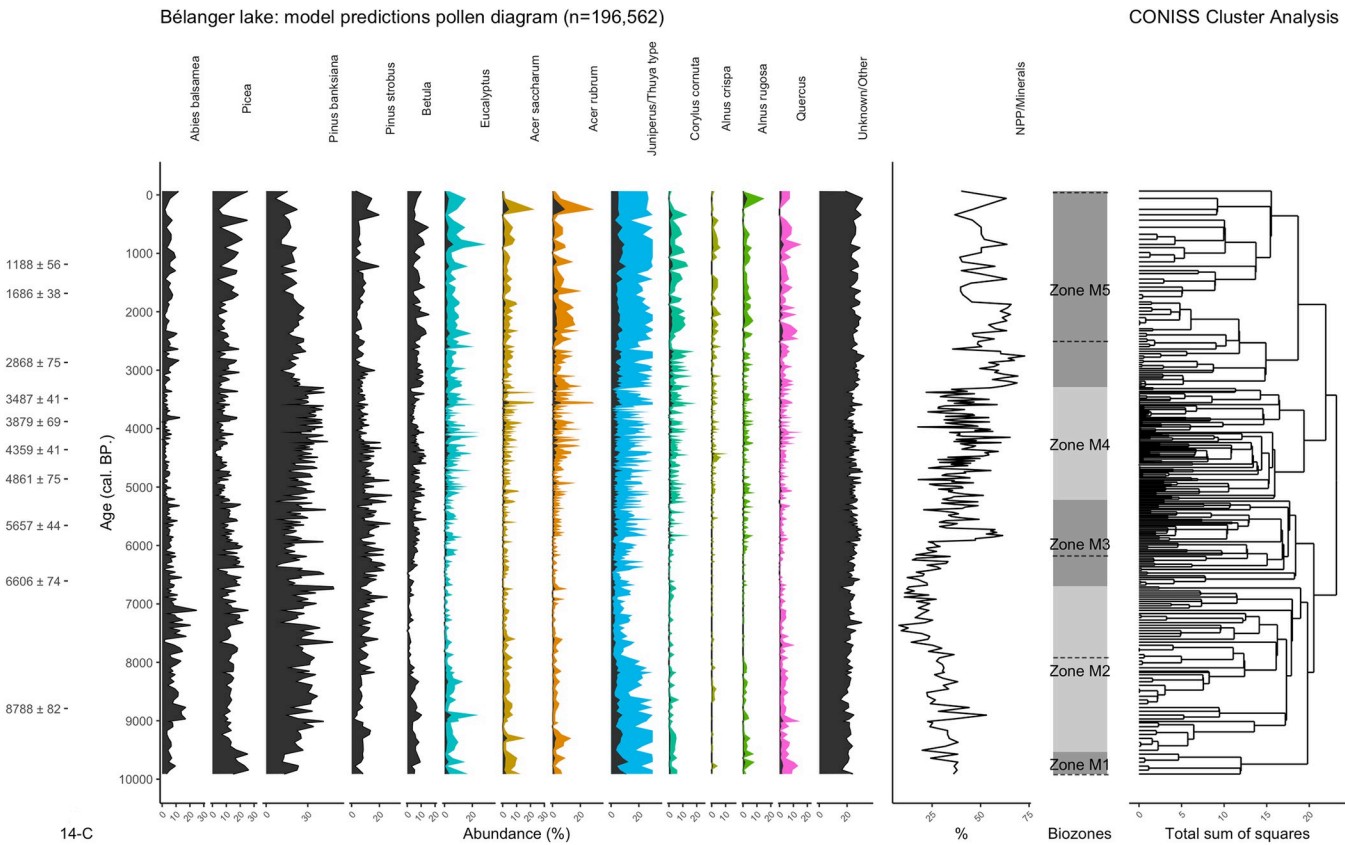

**Fig 8. Pollen abundance diagram plotted using the model's predictions on the full fossil dataset, totalling 271 samples.** When present, the color fills represent a 5x data exaggeration. The total observed amount of Eucalyptus grains was too low to present the data as pollen influx (pollen/cm²/year), so abundance (%) is used instead. The percentage of NPP/minerals per sample is shown as well as the results of the CONISS cluster analysis and its pollen biozones. The dotted lines cutting across the biozones represent the palynologist biozones (Fig 9). The lefthand ticks represent the 14-C dates. See S1 Fig for the age-depth model.

classes' performances–Quercus, Acer ($R^2 = 0.14$; RMSE = 0.01), and Acer rubrum and saccharum.

Both the NPP/minerals and Triporate class ($R^2 = 0.76$; RMSE = 0.02) show a satisfying performance. Their predicted abundance over time closely follows the traditional palynology results. The Betula class indicates a high goodness of fit ($R^2 = 0.73$; 0.02) and likewise shows no clear sign of under- or over-predictions. The Alnus class demonstrates a lower goodness of fit ($R^2 = 0.3$; RMSE < 0.01) yet shows signs of correlation to the palynologist predictions (*p*-value = 0.002). However, two of the Triporate child classes, Eucalyptus and Corylus, show low performance. For Eucalyptus, the model predictions differ from the palynologist predictions mainly in the samples situated at the middle of the time sequence (Fig 6). The Corylus class errors indicate no time-related patterns. Although a pattern of over-prediction can be observed in the Juniperus/Thuja-type class ($R^2 = 0.13$; RMSE = 0.02), its predictions roughly follow the variations of the palynologist predictions.

### 3.3 Comparing the fossil test dataset with the full fossil image dataset

The model was used to classify the images of the full fossil image dataset (Fig 8). 29,760 images fell under the threshold–an average of 14.4%. On average, 38.5% of each slide's images were classified as an NPP/mineral. A second pollen diagram was plotted, representing 30 samples

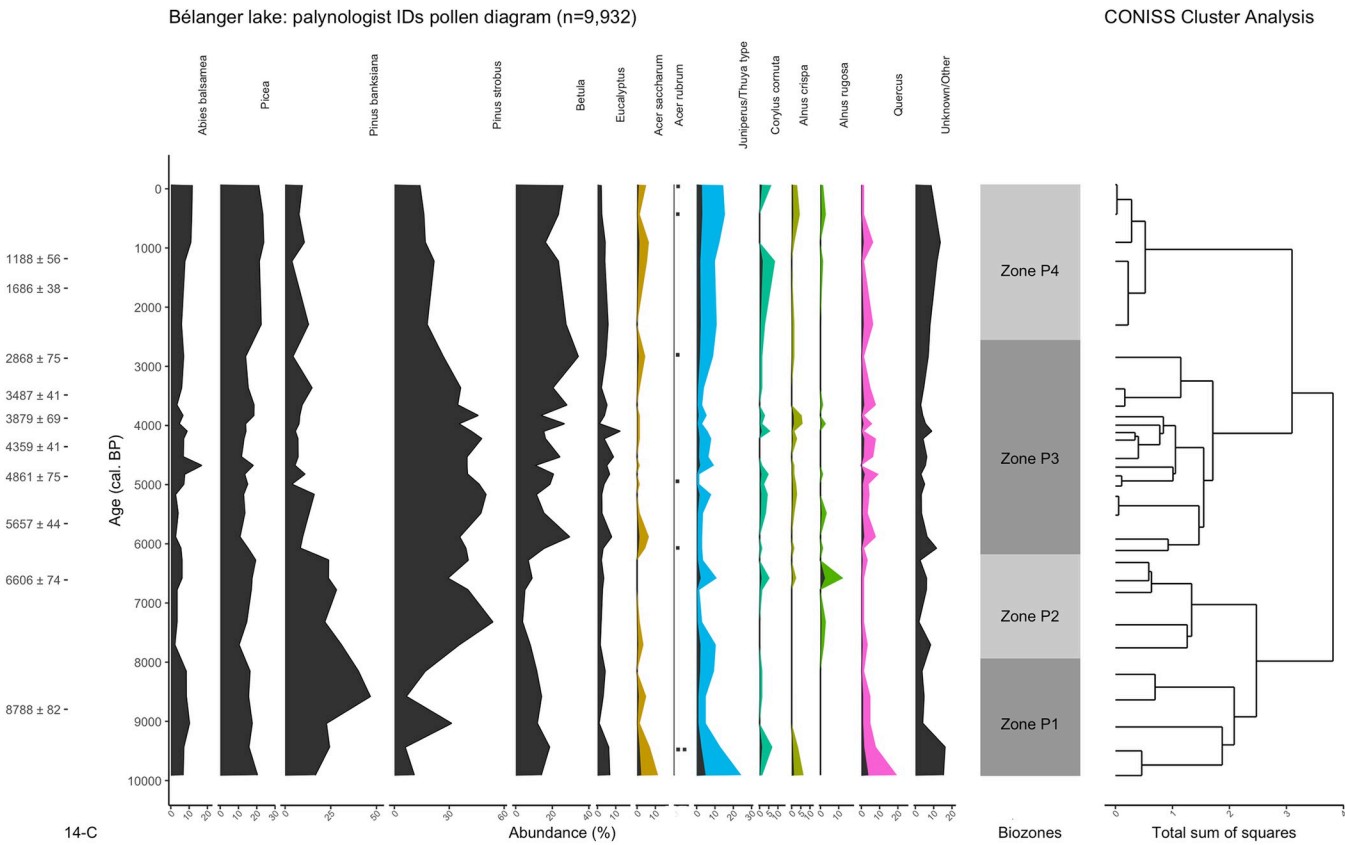

**Fig 9. Pollen abundance diagram plotted using data gathered by a palynologist working on a traditional light microscope.** When present on the diagrams, the color fills represent a 5x data exaggeration. The results of the CONISS cluster analysis and its resulting biozones are also shown. Considering the low *A. rubrum* abundance, its influx is instead represented as dots.

slides from the lac Bélanger fossil sequence whose pollen have been classified by a palynologist using a light microscope (Fig 9)–the traditional method.

There are five distinct pollen biozones in Fig 8 and four in Fig 9. The palynologist diagram's zone P1 (9,900–7,950 cal. BP.) suggests early afforestation patterns characterized by high abundances of *Juniperus communis*, *Picea*, *Betula*, *Quercus* and Unknown/Other pollen. At those latitudes, a high abundance of *Quercus* pollen is often associated with an extra-local influx commonly recorded during periods of low forest density [32]. Similarly, in the early afforestation stages, a significant abundance of Other/Unknown pollen in the fossil record is often associated with the presence of various bushes and other non-ligneous plants [32, 33].

The latter two are extra-local influx commonly recorded during periods of low forest density.

On the model's diagram, this period of afforestation is instead split into two distinct zones. Zone M1 shows the early stages of afforestation characterized by *Quercus* and *Picea* yet fails to register an increase in other unknown extra-local pollen. This further suggests that the model shows overconfidence while identifying pollen belonging to classes it has not been trained on. Zone M2 encloses the later afforestation stage where *P. banksiana* rapidly peaks while *Picea* and *Betula* pollen gradually decrease in abundance. This dynamic is consistent with Fig 9.

On the palynologist diagram, the mid-sequence is grouped into two biozones. Zone P2 (7,950–6,200 cal. BP.) highlights both *Betula*'s lowest abundance and the gradual replacement of *P. banksiana* by *P. strobus*. Zone P3 (6,200–2,550 cal. BP.) is characterized by a progressive

drop in *P. strobus* abundance in favour of *Betula*. The mid-sequence also sees a sporadic influx of less abundant species, namely *A. saccharum* and *Alnus*. These general abundance patterns can also be identified on the model's diagram, where the mid-sequence is similarly characterized by a shift from *P. strobus* to *Betula*. However, the 5,000–3,000 cal. BP. Period showing high *P. banksiana* values is not found on the palynologist diagram. Fig 9 conversely shows that most of the *Pinus* influx consists of *P. strobus*. This is consistent with Fig 6 which shows that the model tends to overestimate the amount of *P. banksiana* and underestimate the amount of *P. strobus*.

Both diagrams show that the top of the sequence (3,000 cal. BP. Onwards) can be grouped in a single zone. Although the model's last zone begins around 800 years earlier, it is consistent with the palynologist diagram's zone P4. In both zones, the taxa diversity of the influx and the absence of a truly dominant taxon can be interpreted as the stabilization of the local vegetation composition The *Pinus* influx is progressively replaced by other pollen from other coniferous species such as *A. balsamea* and *T. occidentalis*. However, the model fails to register an increase in unknown/other pollen as seen on Fig 9.

## 4 Discussion

Our model achieved satisfying results when tested on the test dataset. An APC of 91.2% (with results ranging from 83.3% to 96.6%) is on par with what can be found in the literature. For instance, [13] show an APC between 96.3% and 100%, depending on if their model is classifying intact or fossilised pollen. Another model [22] indicates an APC somewhere between 95.9% and 99.8% while [23] reaches up to 97.9% APC. In [14] the authors train a network on an imbalanced dataset containing 46 different taxa and achieved an 82.3% APC. Their network, trained on 25 species, scored higher at 89.5% APC.

However, we find that our model does not perform as well on the paleo testing dataset classes. A drop in performance was expected on account of the difference in pollen origin (fresh VS fossil) and their respective extraction methods (sections 2.2; 2.4). The classes showing the poorest correlation are the Tricolpate, Acer saccharum, Corylus and Eucalyptus pollen classes, although their low sample count could be a reason for their poor correlation. Regarding the former, we believe that the inadequate correlation could also be due to an elevated presence of non-*Quercus*, non-*Acer* pollen of either Tricolpate or Tricolporate morphology (*e.g. Fagus* or *Tilia* types) in the paleo testing dataset. Although such taxa never reached lac Bélanger's latitudes, their combined abundance in similar pollen records can reach up to 4% on account of long-distance dispersal [32, 34]. Since such pollen were not used to train the model, their recurring presence in the Bélanger fossil record might have been enough to throw off the identification of *Acer* and *Quercus* pollen, thus leading to the over-classification pattern visible in Fig 6. Increasing the prediction threshold to 0.9 results in a higher accuracy for both *Acer* and *Quercus*, further hinting at this conclusion. This would mean that our model could have benefited from being trained on more varied pollen taxa, especially of Tricolpate/Tricolporate morphology.

We find that there is a link between hierarchical level and model performance: the average $R^2$ is higher (= 0.45) for the CNNs situated at the top of the model (Fig 2; CNNs such as Pinus and Triporate) and decreases as the CNNs get deeper ($R^2$ = 0.25 and 0.06 for the mid and bottom level models, respectively). This suggests that compound errors accumulated over two or three successive classifications could result in a drop in performance in the species-level CNNs situated at the bottom of the model.

Certain species see their correlation decline in samples 6,000 cal. BP. And older, such as *A. balsamea*, *Picea* and *Juniperus/Thuja* type (Fig 6). Other species such as *P. banksiana* and

*Eucalyptus* register this drop in correlation only throughout the mid-sequence (6,000–3,000 cal. BP.). We attribute this temporal differentiation in model correlation seen among classes throughout the sequence to differences in the sediments themselves (*e.g.* diagenetic processes varying through time or periods of higher pollen diversity introducing false positives). This is supported by the fluctuating NPP/minerals values seen in Fig 8. The 6,000–3,000 cal. BP. Period averages 42% NPP/minerals abundance (n = 137) while the 9,900–6,000 cal. BP. Period averages 25% (n = 84). An increase in NPP/mineral count signifies an increase in both primary productivity and diagenetic processes, and consequentially a decrease in image and pollen quality. Such dynamics would be less of an issue were the test images manually cropped from a microscope [13], although this would not be adapted to a large-scale workflow.

Both the Triporate and Betula classes achieved a more than satisfying performance on fossil data. Since pollen of the *Betula* genus are typically found in high abundance across North-Eastern American fossil pollen records [32], the model's optimal performance in identifying this class was critical. Similarly, the model's performance in classifying fossil *Pinus* genus pollen is satisfactory, although it appears that the different *Pinus* species could at times be confused with one another (Fig 6). While diagenetic processes explain part of the problem–especially in mid-Holocene samples–we observe that the slide scanner did not pick up on certain discriminating features present in *Pinus* grains. The protruding distal verruca, usually located inbetween the sacci of *P. strobus* grains, are not visible on Fig 3. Visually discriminating morphologically similar pollen grains often depends on precise details, such as the aforementioned distal verruca, a change in texture, or a slight separation of the intine from the exine [3]. While the very large amount of data used in training a CNN can help "brute force" classification, if the quality of the images fail to show such important cues, model performance may be impaired. This could be addressed down the line by using a slide scanner of higher performance, as seen in [24, 35].

Nevertheless, numerous classes show both high training accuracy and good performance on the paleo testing dataset. The NPP/minerals class's high classification accuracy assures us that only a minimal amount of non-pollen palynomorphs was mistakenly classified in other classes. Combined with the rapid laboratory treatment method shown in section 2.4, the high NPP/minerals classification accuracy indicates that a large-scale fossil pollen detection and classification workflow is indeed possible.

By comparing the pollen biozones in Fig 8, we can observe that the variations in the taxa's abundance resemble the ones in the palynologist diagram (Fig 9). While the model is certainly overconfident in its predictions, the general patterns in abundance remain constant with the traditional diagram. The very high temporal resolution allows the addition of a fifth pollen zone in the model's diagram, thus illustrating how an increase in temporal resolution can lead to a more precise sequence. Moreover, the substantial sample size (196,562 images) used in the reconstruction helps mitigating the model's biases, allowing most taxa's abundance patterns to roughly correlate with the traditional diagram. Both the palynologist and model diagrams show abundance patterns that correspond to what can be found in other reconstructions of Québec's mixed-temperate forest region. These patterns, such as the early *Picea* influx or the 7,500–4,000 cal. BP *P. strobus-Betula* dynamic, are observed in regional synthesises such as [32, 36]. The *Juniperus/Thuja* bimodal distribution also corresponds to what is recorded in mixed-temperate forest pollen diagrams–The early influx is commonly associated with *J. communis*, a boreal species [32], while the more recent influx corresponds to *T. occidentalis*, an ubiquist species whose steady influx post 5,000 cal. BP rarely dominates pollen assemblages [36]. Similarly, the re-emergence of *A. balsamea* in the past 3,000 years is recorded in other mixed-temperate pollen diagrams [32, 36].

## 5 Conclusion

In this paper, we presented a large-scale fossil pollen extraction and identification workflow. We proposed a fossil pollen extraction method that is rapid and that leaves the slides clean enough to scan using an automated microscope. We built upon existing pollen classification models to develop our own model. Our model is trained using fresh pollen and is aimed at classifying pollen grains usually found within North-Eastern American Holocene organic sediment records. Our training accuracy is on par with other models currently found in the literature.

However, when tested against the fossil test dataset, its classification accuracy dropped significantly. This drop in accuracy, attributable to the general quality of the fossil pollen and its resulting images, was an expected setback considering that the model had to be trained on fresh pollen. Still, after classifying a full fossil pollen sequence and comparing the results to traditional palynology, we find that our model picks up on most of the general abundance patterns and variations. Furthermore, its results correspond to other pollen diagrams pertaining to similar environments.

As it currently stands, our model is capable of classifying species present in high abundance but fails to classify pollen of rarer species. Such flaws, combined with the fact that there are some less abundant pollen taxa that are still unknown to our model, indicate that additional work should be spent towards bridging the gap between the training data (fresh pollen) and the target data (fossil pollen). In that sense, online repositories of fossil data could prove useful for training providing that data coherence and uniformity are accounted for. Such large repositories would prove beneficial by enabling the training of very broad models whose initial layers could be used in the way of transfer learning. Additionally, given proper class weighting methods, training a model on an imbalanced dataset featuring classes comprised of rarer pollen could prove fruitful. Similarly, the recent publication of a model adapted to temperate European taxa points towards the fact that training on a very small, carefully assembled fossil dataset can yield very promising results [11]. Nevertheless, combined with an automated slide scanner and our accelerated pollen extraction method, our model serves as a proof of concept for the implementation of a fully automatic large-scale fossil pollen classification workflow.

## Supporting information

**S1 Fig. Bélanger lake age/depth model curve.** 10 samples of bulk gittya bulk were dated. The blue lines represent the samples' age and their confidence error. The age/depth model was achieved using a smooth-spline function.
(JPEG)

**S1 File. Classifynder image capture parameters.** These are the parameters input into the Classifynder automated slide scanner software while gathering data from slides.
(TXT)

## Acknowledgments

First, we wish to acknowledge that lac Bélanger is located in Nitaskinan. It is unceded indigenous land that has traditionally been under the guardianship of the Atikamekw Nehirowisiwok people and upon which they still live to this day. The authors wish to thank A. Hennebelle and L. Peter for sampling the lac in 2017. Special thanks to D. Belamy for technical contributions, to Lucas CA for help on Figs 1 and 4, and to J. Aleman for overall support and guidance in the early stages of the project. Finally, the authors wish to thank the Ministère des Ressources naturelles et des Forêts du Québec (MRNF) for their help given along the project, as well as H.

John. B. Birks for their constructive insight during the reviewing process. Finally, we thank T. Iaculli for proofreading the manuscript and V. Poirier (MRNF) for creating the map.

## Author Contributions

**Conceptualization:** Médéric Durand, Olivier Blarquez.

**Data curation:** Médéric Durand, Jordan Paillard.

**Formal analysis:** Médéric Durand.

**Funding acquisition:** Pierre Grondin, Olivier Blarquez.

**Investigation:** Médéric Durand.

**Methodology:** Médéric Durand, Jordan Paillard, Marie-Pier Ménard.

**Project administration:** Pierre Grondin, Olivier Blarquez.

**Resources:** Jordan Paillard, Olivier Blarquez.

**Software:** Médéric Durand.

**Supervision:** Pierre Grondin, Olivier Blarquez.

**Validation:** Médéric Durand.

**Visualization:** Médéric Durand.

**Writing – original draft:** Médéric Durand.

**Writing – review & editing:** Médéric Durand, Thomas Suranyi, Pierre Grondin, Olivier Blarquez.

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
