## [Decision Letter · Decision Letter 0]

18 Jul 2023

PONE-D-23-19374Pollen identification through convolutional neural networks: first application on a full fossil pollen sequencePLOS ONE

Dear Dr. Durand,

Thank you for submitting your manuscript to PLOS ONE. After careful consideration, we feel that it has merit but does not fully meet PLOS ONE’s publication criteria as it currently stands. Therefore, we invite you to submit a revised version of the manuscript that addresses the points raised during the review process.

We look forward to receiving your revised manuscript.

Kind regards,

Xiaoyong Sun

Academic Editor

PLOS ONE

Journal Requirements:

3. We note that Figure 1 in your submission contain map/satellite images which may be copyrighted. All PLOS content is published under the Creative Commons Attribution License (CC BY 4.0), which means that the manuscript, images, and Supporting Information files will be freely available online, and any third party is permitted to access, download, copy, distribute, and use these materials in any way, even commercially, with proper attribution. For these reasons, we cannot publish previously copyrighted maps or satellite images created using proprietary data, such as Google software (Google Maps, Street View, and Earth). For more information, see our copyright guidelines: http://journals.plos.org/plosone/s/licenses-and-copyright.

Reviewers' comments:

Reviewer's Responses to Questions

**Comments to the Author**

1. Is the manuscript technically sound, and do the data support the conclusions?

Reviewer #1: Yes

Reviewer #2: Partly

2. Has the statistical analysis been performed appropriately and rigorously? 

Reviewer #1: No

Reviewer #2: Yes

3. Have the authors made all data underlying the findings in their manuscript fully available?

Reviewer #1: No

Reviewer #2: Yes

4. Is the manuscript presented in an intelligible fashion and written in standard English?

Reviewer #1: Yes

Reviewer #2: Yes

5. Review Comments to the Author

Reviewer #1: This is the review report of the paper which is titled “

Pollen identification through convolutional neural networks: first application on a full fossil pollen sequence “.

The paper required more effort to be ready.

1- The abstract lacks open questions and limitations of the AI existing methods.

2- Figure 2 is the main one is fully unclear and not able to judge it. Please redraw it.

3- Contributions of the paper should be listed at the end of introduction.

4- Lack of statistical evaluation. Please list them as a table

5- Paper presentation has to be approved Such as adding Grad-Cam.

6- list the issues with previous methods that this paper solved in related work.

7- Comparison with recent previous methods (2021-2022-2023) in terms of results on same used datasets with proper citation of previous methods. I would suggest validating the proposed solution on more than one public dataset.

Reviewer #2: The study applies nested machine learning models to the identification of Holocene pollen samples from Lac Berlanger, Quebec. The study ambitiously attempts to transfer learning from reference/fresh pollen images to fossil samples and is an application of the method presented in Bourel et al 2020 (citation #7). The machine counts, however, do not closely match the human palynologist counts.

The main issues with the paper stem from some of the choices made in the construction of their CNN models. More explanation/detail is needed.

For example, it is unclear why choosing to "transformed to grayscale and rescaled to 128x128 pixels" (Line 197-8) would standardize the data and improve training performance. This more likely prevented further improvement using RGB colors and larger resolutions. It would be better to just say "we transformed ... in this work".

The choice to limit each CNN's corresponding dataset size to the number of images present in least populated class" (Line 221), throws away a large amount of valuable data. Training on imbalanced data is an established topic in computer vision and there are alternate approaches. A more natural solution would be to adopt weighted loss which assigns a larger weight to rare class.

Lines 241-246 mentions "transfer learning" but this is actually a discussion of network architecture. The authors should explain how the network was pretrained and on what data, such that the weights can be transferred to pollen data.

The choice to train multiple CNNs was unusual – including the use of a shallow network to train the Alnus CNN because "(deeper models) tend to overfit rapidly" (Line 268). Why did the authors not train a single CNN backbone shared by all classes using all data (allowing learning richer features for all classes)?

Overall, the machine counts did not appear to closely track human/manual counts. The authors make two interesting choices in their comparison. First – human counts are made from images and not from a microscope. This should remove the effect of poor-quality images, as both human and machine classifications would be affected. But there is still low correlation between the two counts outside of a handful of taxa, e.g. Pinus (Figure 7). Having the human analysis identify detected grains/cropped images doesn’t test whether all pollen are identified by the Classifynder, or if there is any systematic bias in what is automatically imaged/detected. Second – given that this is a proof-of-concept paper, why not manually count all 271 samples so that the manual and automatic counts and zonation could be more accurately compared?

Based on the images in Figure 3, the discussion of image quality (Lines 526-9) should be significantly expanded. It may be the quality of the images that are the issue and not your models. See the following papers on what is possible with higher quality images:

Punyasena, S. W., Tcheng, D. K., Wesseln, C., & Mueller, P. G. (2012). Classifying black and white spruce pollen using layered machine learning. New Phytologist, 196(3), 937-944.

Kong, S., Punyasena, S., & Fowlkes, C. (2016). Spatially aware dictionary learning and coding for fossil pollen identification. In Proceedings of the IEEE Conference on Computer Vision and Pattern Recognition Workshops (pp. 1-10).

Romero, I. C., Kong, S., Fowlkes, C. C., Jaramillo, C., Urban, M. A., Oboh-Ikuenobe, F., ... & Punyasena, S. W. (2020). Improving the taxonomy of fossil pollen using convolutional neural networks and superresolution microscopy. Proceedings of the National Academy of Sciences, 117(45), 28496-28505.

Additional comments:

L 55: Mander et al 2014 (citation #4) focuses on the classification of grass pollen, which is notoriously difficult to tell apart, even under electron microscopy. It is not the best example of the reproducibility of visual pollen identification and puts traditional palynological data collection in a more negative light than necessary.

L 56: McLeod et al 2010 does not focus on palynology. It also predates the bulk of machine learning research in palynology. There are better references that would characterize the current state of palynological research.

L 65: A number of papers have discussed slide scanning and can be cited here, including:

Holt, K. A., & Bebbington, M. S. (2014). Separating morphologically similar pollen types using basic shape features from digital images: A preliminary study. Applications in Plant Sciences, 2(8), 1400032.

L 70, L 90-95: Punyasena et al 2022 (citation #8) also used a slide scanner and image segmentation/pollen detection instead of manual crops. While these were environmental pollen samples (from pollen traps), they have similar issues to Quaternary pollen samples.

L 173: Were vouchers collected to confirm and record species identifications?

L 190: Provide more detail on how “pollen was automatically identified through the use of morphological criteria input by the user”. Is this using the Classyfinder software? If so, explain how this is automated and what specific criteria are used.

L 192: Are you using a Z-stack (multiple images) of pollen specimen or a single fused image? This is not clear.

L 208: The GitHub repository mentions that the Python code should work with PyTorch. Might be worth mentioning that here too.

L 241: What is meant by the “bottom” of the model? The initial layers?

L 299: Using different processing methods for reference pollen, fresh pollen, and fossil pollen introduces visual artifacts that makes the classification problem much harder than it should be. Ideally, as many variables as possible should be controlled. The authors should consider comparing like with like in their training and testing and explicitly test whether processing choices affected the results.

L 326: As per the comment above, any visual documentation of the differences in this alternate processing method?

L 553: The word “recrudescence” has negative connotations. Use “re-emergence” or “reappearance” instead.

Recurring: Recommend using a spelling/grammar check to catch errors. Species and genera names should be italicized.

Figure 3: Include example images of all 13 classes.

Figures 8 and 9: It would be helpful to see zonations overlaid or side by side.

6. PLOS authors have the option to publish the peer review history of their article (what does this mean?). If published, this will include your full peer review and any attached files.

Reviewer #1: **Yes: **Laith Alzubaidi

Reviewer #2: No

---

## [Author Response · Author response to Decision Letter 0]

5 Dec 2023

Pollen identification through convolutional neural networks: First application on a full fossil pollen sequence

Review comments to the authors

Comments from the Academic Editor

Please ensure that your manuscript meets PLOS ONE’s style requirements, including those for file naming.

- The revised manuscript now meets the style requirements. In particular, this is in regard to file naming, SI naming, the bibliography and the heading font size. 

We note that the grant information you provided in the ‘Funding Information’ and ‘Financial Disclosure’ sections do not match. 

- As established in the additional comments section of the Submission system: “The funding acquired through the Ministère des Ressources naturelles et des forêts (Québec) has been attributed in your submission system to the "Ministère des Ressources naturelles et de l'énergie". The latter does not exist anymore, but we entered it as such because the former does not exist within your system as of yet.”

We note that Figure 1 in your submission contain map/satellite images which may be copyrighted. 

- The map in figure 1 has been created by a credited team member (Véronique Poirier) working for the Québec provincial government. She uses data available through the provincial government databases – contour lines and forest zones, as well as waterbodies. The database is called Forêt-Ouverte (open forest), more info can be found here (In English): https://mffp.gouv.qc.ca/documents/forest/user_guide_foret_ouverte.pdf

- And here, in French, which states that the database is under a CC-BY 4.0 Licence: 

https://www.donneesquebec.ca/recherche/dataset/carte-ecoforestiere-avec-perturbations/resource/7ada89ee-0d80-4c6a-b4e3-1090b18e34b8

- Many papers have been published featuring maps that use the same governmental databases without any copyright issues. Pierre Grondin, who just in case has signed the copyright form allowing for use of this data, is a researcher for the Natural Ressources and Forest Ministry of Québec (MRNF).

 

Comments from Reviewer #1:

The abstract lacks open questions and limitations of the AI existing methods.

- An additional line concerning a limitation of AI methods in fossil palynology has been added to the revised manuscript. We otherwise believe limitations concerning existing AI methods are already listed in the Abstract: “Still, only a small portion of works published on the matter address the classification of fossil pollens”; “[…] there exists a gap between the training data and the target data”.

Figure 2 is the main one is fully unclear and not able to judge it. Please redraw it.

- Unfortunately, we do not see how to more efficiently convey the information presented in Figure 2. This figure allows the reader to understand the hierarchical aspect of our method. We believe the color coding and legend help the readers to work their way around the figure. The white boxes give relevant information on the CNNs’ parameters for those willing to play around the model. The blue boxes give important information to paleoecologists versed in Northeastern American pollens interested in knowing which species were used in forming the initial training classes.

- We tested moving the white boxes into a separate table, but they are hardly separable – understanding the content of the white boxes requires seeing the tree-like structure and how the CNNs relate to one another. Since this figure is referred to many times throughout the manuscript, this slowed down the reading process.

- We did, however, add a bit more meat onto the figure description in order to further guide the reader.

Contributions of the paper should be listed at the end of introduction.

- This has been added to the revised manuscript.

Lack of statistical evaluation. Please list them as a table

- A table has been added to the revised manuscript (Table 3)

Paper presentation has to be approved Such as adding Grad-Cam.

- We believe that adding Grad-cam visualization would burden an already elaborate and multi-faceted manuscript. Grad-cam is seldom used in other papers pertaining to pollen classification networks.

List the issues with previous methods that this paper solved in related work.

- The performance of our method is compared to the performance of other similar methods in the discussion. However, since our method is the first of its kind to be trained on pollens found in Quebec organic fossil records, it cannot directly solve issues found in related works as: a) their pollen data come from plant assemblages not endemic to Quebec; and b) no other paper – that we know of – has tackled the challenge of classifying pollens found in organic fossil records. We emphasize the fact that the goal of this manuscript is not so much displaying new methods in pollen classification networks as it is creating a freely available tool for palynologists working on Quebec fossil pollen records. We do, however, list the issues with previous methods in traditional (Non-AI) palynology in the abstract and introduction sections.

Comparison with recent previous methods (2021-2022-2023) in terms of results on same used datasets with proper citation of previous methods. I would suggest validating the proposed solution on more than one public dataset.

- Comparing our method and its results to other published methods is not possible. This is because our method is only able to classify the pollens and objects most commonly found in Quebec organic sediment records. In other words, no public models can classify our target pollens and we cannot classify the target data of other pollen classification models. No labelled dataset suited for our topic currently exist. This is the reason why we had to resort to validating our results using human palynologist predictions (a time intensive process that was necessary in order to achieve any kind of data validation). We thus cannot validate our method on any other public dataset or compare it any further than we already have on recent published methods.

Comments from reviewer #2:

For example, it is unclear why choosing to "transformed to grayscale and rescaled to 128x128 pixels" (Line 197-8) would standardize the data and improve training performance. This more likely prevented further improvement using RGB colors and larger resolutions. It would be better to just say "we transformed ... in this work".

- This has been changed in the revised manuscript.

The choice to limit each CNN's corresponding dataset size to the number of images present in least populated class" (Line 221), throws away a large amount of valuable data. Training on imbalanced data is an established topic in computer vision and there are alternate approaches. A more natural solution would be to adopt weighted loss which assigns a larger weight to rare class.

- We recognize that neglecting to train the data on imbalanced data was an oversight – we then believed that diving into weighted loss methods would fall outside of the scope of the paper. This is now addressed in the conclusion of the revised manuscript, and will be identified as something to add onto future work. 

Lines 241-246 mentions "transfer learning" but this is actually a discussion of network architecture. The authors should explain how the network was pretrained and on what data, such that the weights can be transferred to pollen data.

- We included more information on VGG16 and the ImageNet dataset in the revised manuscript. In the quoted paragraph, a few lines already establish how VGG16 can help our own model extract basic features from our data, with references to other papers in automated palynology. We added a line to further explain this, as well as a reference to Alzubaidi et al. (2021).

The choice to train multiple CNNs was unusual – including the use of a shallow network to train the Alnus CNN because "(deeper models) tend to overfit rapidly" (Line 268). Why did the authors not train a single CNN backbone shared by all classes using all data (allowing learning richer features for all classes)?

- The inherent taxonomic nature of pollens allowed us to work in a hierarchical classification system (as seen in figure 2). This permits us to have a pollen drop out of the classification and be classified as at the genus level if it falls under a confidence threshold, as seen in Bourel et al. (2020). Similarly to their work, we trained multiple, smaller, CNNs instead of a single one.

- Training a single CNN would have us face the issue of an important data imbalance (further addressed in the now revised conclusion) since certain species-level classes had ten times the amount of training data than some others. On the other hand, this allowed us to capitalize on the large amount of training data available when grouping certain species in taxonomic classes. For instance, the top classifier trained on 1,200 images per class. 

- As proposed in the reviewer comment, something that could have been done is using a CNN trained on all classes and using its base layers in transfer learning. This is now addressed in the revised conclusion.

Overall, the machine counts did not appear to closely track human/manual counts. The authors make two interesting choices in their comparison. First – human counts are made from images and not from a microscope. This should remove the effect of poor-quality images, as both human and machine classifications would be affected. But there is still low correlation between the two counts outside of a handful of taxa, e.g. Pinus (Figure 7). Having the human analysis identify detected grains/cropped images doesn’t test whether all pollen are identified by the Classifynder, or if there is any systematic bias in what is automatically imaged/detected. Second – given that this is a proof-of-concept paper, why not manually count all 271 samples so that the manual and automatic counts and zonation could be more accurately compared?

- We indeed chose to validate our model’s results using images captured by the Classyfinder microscope. This is addressed in the original manuscript (line 392). In short, this was the only way to isolate our model’s own error and bias. What we propose in this paper is not a fully integrated system that requires the Classifynder microscope. Rather, it is a classification tool fit for any fossil pollen sequence whose images have been digitized using any form of light-microscope, automated or not. 

- Second – while this would have improved the proof-of-concept aspect of our manuscript, counting the full 271 samples would imply counting and identifying a little under 200,000 images. A large part of those images do not belong to any of our trained classes – a single sample can often contain pollen grains and spores from more than 30 species. Since pollen identification is a tedious process and requires a trained specialist, this fell outside of the time and financial limits of the project. 

Based on the images in Figure 3, the discussion of image quality (Lines 526-9) should be significantly expanded. It may be the quality of the images that are the issue and not your models. See the following papers on what is possible with higher quality images:

Punyasena, S. W., Tcheng, D. K., Wesseln, C., & Mueller, P. G. (2012). Classifying black and white spruce pollen using layered machine learning. New Phytologist, 196(3), 937-944.

Kong, S., Punyasena, S., & Fowlkes, C. (2016). Spatially aware dictionary learning and coding for fossil pollen identification. In Proceedings of the IEEE Conference on Computer Vision and Pattern Recognition Workshops (pp. 1-10).

Romero, I. C., Kong, S., Fowlkes, C. C., Jaramillo, C., Urban, M. A., Oboh-Ikuenobe, F., ... & Punyasena, S. W. (2020). Improving the taxonomy of fossil pollen using convolutional neural networks and superresolution microscopy. Proceedings of the National Academy of Sciences, 117(45), 28496-28505.

- We are thrilled to see how super high-resolution confocal/fluorescence imaging can be integrated into deep learning methods with such good results. We wonder if the above mentioned methods using Zeiss products would be operable for high-volume imaging? Nonetheless, this further shows how imaging may have had a bigger impact on our model than previously thought. We have expanded the discussion of image quality in the revised manuscript.

L 55: Mander et al 2014 (citation #4) focuses on the classification of grass pollen, which is notoriously difficult to tell apart, even under electron microscopy. It is not the best example of the reproducibility of visual pollen identification and puts traditional palynological data collection in a more negative light than necessary.

- This is a valid concern. We revised the manuscript in consequence, and referenced other works that touch on the general coherency between analysts.

L 56: McLeod et al 2010 does not focus on palynology. It also predates the bulk of machine learning research in palynology. There are better references that would characterize the current state of palynological research.

- We revised the manuscript in consequence; Holt & Bennet (2014) and Mander & Punyasena (2018) give much more accurate insight on the shortcomings of pollen analysis in their respective review papers. Both identify the possible benefits of automated palynology. 

L 65: A number of papers have discussed slide scanning and can be cited here, including:

Holt, K. A., & Bebbington, M. S. (2014). Separating morphologically similar pollen types using basic shape features from digital images: A preliminary study. Applications in Plant Sciences, 2(8), 1400032.

- In the revised manuscript, cited works related to introducing a deep-learning/slide-scanner workflow, including Holt and Bebbington (2014), Sevillano, Holt & Aznarte (2020) and Pedersen et al (2017).

L 70, L 90-95: Punyasena et al 2022 (citation #8) also used a slide scanner and image segmentation/pollen detection instead of manual crops. While these were environmental pollen samples (from pollen traps), they have similar issues to Quaternary pollen samples.

- L70: Added Punyasena et al 2022 to the other citations.

- L90-95: Cited Punyasena et al 2022 as an example of combining image detection and an automated slide scanner in the revised manuscript.

L 173: Were vouchers collected to confirm and record species identifications?

- We are regretfully unsure of the meaning of this question. The chosen tree species are handily identifiable in the wild and are not known to present a challenge in the matter.

L 190: Provide more detail on how “pollen was automatically identified through the use of morphological criteria input by the user”. Is this using the Classyfinder software? If so, explain how this is automated and what specific criteria are used.

- This was using the Classyfinder software. We added the specific criteria in the SI.

L 192: Are you using a Z-stack (multiple images) of pollen specimen or a single fused image? This is not clear.

- This was using a single fused image. Line has been reworked to reflect this in the revised manuscript.

L 208: The GitHub repository mentions that the Python code should work with PyTorch. Might be worth mentioning that here too.

- Added to the revised manuscript.

L 241: What is meant by the “bottom” of the model? The initial layers?

- The initial layers. Line has been revised in the new manuscript.

L 299: Using different processing methods for reference pollen, fresh pollen, and fossil pollen introduces visual artifacts that makes the classification problem much harder than it should be. Ideally, as many variables as possible should be controlled. The authors should consider comparing like with like in their training and testing and explicitly test whether processing choices affected the results.

- There is regretfully no way around processing the fresh and fossil pollens differently. This is addressed in section 2.4, as well as what laboratory methods were used to minimize discrepancy. The goal of this project was to create a model capable of classifying fossil pollens extracted from a sediment core. The fossil pollens unavoidably undergo a different processing method than fresh pollens do. This is mostl

---

## [Decision Letter · Decision Letter 1]

8 Jan 2024

PONE-D-23-19374R1Pollen identification through convolutional neural networks: first application on a full fossil pollen sequencePLOS ONE

Dear Dr. Durand,

Thank you for submitting your manuscript to PLOS ONE. After careful consideration, we feel that it has merit but does not fully meet PLOS ONE’s publication criteria as it currently stands. Therefore, we invite you to submit a revised version of the manuscript that addresses the points raised during the review process. As you can see the comments of the reviewers, you need to address all questions of reviewers before we can consider the next step.

We look forward to receiving your revised manuscript.

Kind regards,

Xiaoyong Sun

Academic Editor

PLOS ONE

Reviewers' comments:

Reviewer's Responses to Questions

**Comments to the Author**

1. If the authors have adequately addressed your comments raised in a previous round of review and you feel that this manuscript is now acceptable for publication, you may indicate that here to bypass the “Comments to the Author” section, enter your conflict of interest statement in the “Confidential to Editor” section, and submit your "Accept" recommendation.

Reviewer #1: (No Response)

Reviewer #3: (No Response)

2. Is the manuscript technically sound, and do the data support the conclusions?

Reviewer #1: No

Reviewer #3: Yes

3. Has the statistical analysis been performed appropriately and rigorously? 

Reviewer #1: Yes

Reviewer #3: Yes

4. Have the authors made all data underlying the findings in their manuscript fully available?

Reviewer #1: Yes

Reviewer #3: Yes

5. Is the manuscript presented in an intelligible fashion and written in standard English?

Reviewer #1: (No Response)

Reviewer #3: Yes

6. Review Comments to the Author

Reviewer #1: The authors did not put enough effort to improve the paper

The comments haven't addreesed very well, for example

1- The quilty of figure 2 very poor, I can't read what was written

2- it is not clear what is the novely of the paper

3- please go back to main comments and address them.

Reviewer #3: l 21,43 not only environments, but flora & vegetation

l 54,56 consistency, not consistence

l 55 All the more - unclear meaning

l 61 ecological processes are inferred, not observed

l 75.76 I think [11-14] is the subject in this sentence

1 100 full stop needed, not-

l 155 taxa, not species

Table 2 Thuja, not Thuya

l 169 and throughout pollen, not pollens, Pollen, not Pollens

l 179 replace the first P. with Picea, and the third P. with Pinus

l 211, 213 is should be are as data are plural

l 212 'its' should be 'their'

l 218 similarity, not similitude

l 222 add 'the' before least

l 229 delete of

l 287 precipitation, not precipitations

l 288 Abies, not A.

l 289 Alnus, not A.

l 308 16,319 +-1,975 - is this numbers per cc?

l 319 centrifuged, not centrifugated

l 333 broken-stick is a means of identifying how many zones should (from CONISS or other zonation methods) to be adopted.

l 340 could not, not couldn't

l 508 Pinus, not P.

l 503 recrudescence - is this a word?

l 597,598 This should be Birks, HJB & Birks, HH Quaternary Palaeoecology Edward Arnold, London. I am unaware of any connection with the University of California Press.

7. PLOS authors have the option to publish the peer review history of their article (what does this mean?). If published, this will include your full peer review and any attached files.

Reviewer #1: No

Reviewer #3: **Yes: **H. John B. Birks

---

## [Author Response · Author response to Decision Letter 1]

7 Feb 2024

Please refer to the last section of the document for answers to specific reviewer comments.

Concerning Figure 2: We fail to interpret Reviewer's #1 comment concerning figure 2. If by 'fully unreadable' it was meant that the figure is confused and difficult to interpret, know that the figure has been redrawn and simplified. Otherwise, if it was meant that the figure was literally unreadable because of poor quality, know that this is an effect of Plos' editorial manager file compression system. The original figure is not compressed and is fully readable.

---

## [Decision Letter · Decision Letter 2]

21 Feb 2024

PONE-D-23-19374R2Pollen identification through convolutional neural networks: first application on a full fossil pollen sequencePLOS ONE

Dear Dr. Durand,

Thank you for submitting your manuscript to PLOS ONE. After careful consideration, we feel that it has merit but does not fully meet PLOS ONE’s publication criteria as it currently stands. Therefore, we invite you to submit a revised version of the manuscript that addresses the points raised during the review process.

We look forward to receiving your revised manuscript.

Kind regards,

Xiaoyong Sun

Academic Editor

PLOS ONE

Journal Requirements:

Reviewers' comments:

Reviewer's Responses to Questions

**Comments to the Author**

1. If the authors have adequately addressed your comments raised in a previous round of review and you feel that this manuscript is now acceptable for publication, you may indicate that here to bypass the “Comments to the Author” section, enter your conflict of interest statement in the “Confidential to Editor” section, and submit your "Accept" recommendation.

Reviewer #1: All comments have been addressed

Reviewer #3: All comments have been addressed

2. Is the manuscript technically sound, and do the data support the conclusions?

Reviewer #1: Yes

Reviewer #3: Yes

3. Has the statistical analysis been performed appropriately and rigorously? 

Reviewer #1: Yes

Reviewer #3: Yes

4. Have the authors made all data underlying the findings in their manuscript fully available?

Reviewer #1: Yes

Reviewer #3: Yes

5. Is the manuscript presented in an intelligible fashion and written in standard English?

Reviewer #1: Yes

Reviewer #3: Yes

6. Review Comments to the Author

Reviewer #1: The authors a]have addressed all of the comments except Figure 2 is not clear, it is hard to read it.

Reviewer #3: See attached file that I have sent. It is a docx file. I recommend that the manuscript be accepted subject to some minor revisions.

7. PLOS authors have the option to publish the peer review history of their article (what does this mean?). If published, this will include your full peer review and any attached files.

Reviewer #1: No

Reviewer #3: **Yes: **H. John B. Birks

---

## [Author Response · Author response to Decision Letter 2]

6 Mar 2024

Please refer to the last section of the document for answers to specific reviewer

comments.

Concerning Figure 2: It appears that fig. 2 is of poor quality in the document that is sent to the reviewers. This is most likely a result of the compression from the editorial manager. The original figure is not compressed, and is perfectly readable. In the Manuscript Draft, you may click on fig2 to see the full resolution figure.

---

## [Editor Report · Decision Letter 3]

17 Mar 2024

PONE-D-23-19374R3Pollen identification through convolutional neural networks: first application on a full fossil pollen sequencePLOS ONE

Dear Dr. Durand,

Thank you for submitting your manuscript to PLOS ONE. After careful consideration, we feel that it has merit but does not fully meet PLOS ONE’s publication criteria as it currently stands. Therefore, we invite you to submit a revised version of the manuscript that addresses the points raised during the review process.

We look forward to receiving your revised manuscript.

Kind regards,

Xiaoyong Sun

Academic Editor

PLOS ONE
---

## [Author Response · Author response to Decision Letter 3]

20 Mar 2024

Please refer to the last section of the document for answers to specific reviewer

comments.

Concerning Figure 2: It appears that fig. 2 is of poor quality in the document that is sent

to the reviewers. This is most likely a result of the compression from the editorial

manager. The original figure is not compressed, and is perfectly readable. In the

Manuscript Draft, you may click on fig2 to see the full resolution figure.

---

## [Editor Report · Decision Letter 4]

4 Apr 2024

Pollen identification through convolutional neural networks: first application on a full fossil pollen sequence

PONE-D-23-19374R4

Dear Dr. Durand,

We’re pleased to inform you that your manuscript has been judged scientifically suitable for publication and will be formally accepted for publication once it meets all outstanding technical requirements.

Kind regards,

Xiaoyong Sun

Academic Editor

PLOS ONE